# Inconsistencies between observed, reconstructed, and simulated precipitation indices for England since the year 1650 CE

Oliver Bothe[1], Sebastian Wagner[1], and Eduardo Zorita[1]

[1]Helmholtz Zentrum Geesthacht, Institute of Coastal Research, 21502 Geesthacht, Germany

**Correspondence:** Oliver Bothe (ol.bothe@gmail.com)

**Abstract.** The scarcity of long instrumental records, uncertainty in reconstructions, and insufficient skill in model simulations hamper assessing how regional precipitation changed over past centuries. Here, we use standardised precipitation data to compare a regional climate simulation, reconstructions, and long observational records of seasonal (March to July) mean precipitation in England and Wales over the past 350 years. The Standardized Precipitation Index is a valuable tool for bridging part of the problems in assessing agreement and disagreement between the different sources of information. We assess the agreement in the temporal evolution of percentiles of the precipitation distributions. These evolutions are not consistent among reconstructions, a regional simulation, and instrumental observations for severe and extreme dry and wet conditions. The lack of consistent relations between the different data sets may be due to the dominance of internal climate variability over the purely natural exogenous forcing conditions on multi-decadal time-scales. This, in turn, questions our ability to make dynamical inferences about hydroclimate variability for small regions. However, it is encouraging that there is still some agreement between a regional simulation and observational indices. Our results emphasize the complexity of hydroclimate changes during the most recent historical period and stress the necessity of a thorough understanding of the processes affecting forced and unforced precipitation variability.

## 1 Introduction

Confidence in future climate projections of, e.g., drought and wetness conditions requires understanding of past climate and hydroclimate variability and its drivers (e.g. Schmidt et al., 2014). Focussing on the hydroclimate, estimates of past and future changes are still highly uncertain for precipitation at regional scales. Indeed, our understanding of internal, naturally forced, and anthropogenically forced variability is weaker for precipitation than for temperature due to the more complex controls on precipitation variability (e.g. Zhang et al., 2007; Hoerling et al., 2009; Iles et al., 2013; Fischer et al., 2014) and the more local-scale nature of precipitation-processes.

Consistency among estimates from early instrumental observations, paleo-reconstructions from environmental archives (i.e., paleo-observations), and climate simulations supports our understanding of past changes. Here, consistency among estimates simply means that various sources of information do not contradict each other. Consistency is a weaker requirement than to expect consilience, i.e., it is weaker than requiring that the evidence from the different data sets converges. Despite being a

more liberal metric, consistency is an appropriate measure in view of the multiple sources of uncertainty in inferring past hydroclimate and precipitation variability.

Here, we explore consistency and inconsistency of observations, reconstructions, and simulations for one small region and focusing only on precipitation changes. Specifically we set out to study the consistency in the statistical properties of precipitation distributions in these sources of information.

Comparing precipitation among different data sources poses various challenges. Problems relate not only to pronounced biases in the simulated precipitation, especially derived from raw global models, and to differences in representation or, in the case of data fields, the grid resolution. In the context of long observational time series, data inhomogeneities due to changes in instrumentation, measuring techniques, and changes in locations can further influence estimates of longer-term trends (Frank et al., 2007; Wilson et al., 2005; Böhm et al., 2010). Reconstructions likely represent only part of the variability spectrum of, e.g., precipitation, dependent on the strength of the climatic signal in the original data and on further shortcomings of the underlying paleo-observations (compare discussions by PAGES Hydro2k Consortium, 2017). The PAGES Hydro2k Consortium (2017) discuss in more detail the problems in comparing hydroclimatic variables between reconstructions and simulations.

The PAGES Hydro2k Consortium (2017) developed recommendations for the comparison of hydroclimate representations in simulations and paleo-observations, emphasizing the uncertainties of estimates from both sources. They stress the complementary nature of simulated and environmental information. Estimates have to represent the same parameters on related spatial and temporal scales. Only then, a comparison can be valid. We need appropriate methods to bridge the gap between the local or regional reconstruction and the simulation output that represents aggregates over larger spatial scales. Proxy system models are one means to achieve this (Evans et al., 2013; PAGES Hydro2k Consortium, 2017). We argue that the standardisation of precipitation estimates is a simple means to compare the statistical properties of hydroclimatic parameters in simulations and paleo-observations complementing the current suite of statistical diagnostics for model-data comparisons. It is of value for periods with and without comprehensive sets of climate and weather observations.

Transforming precipitation estimates to the Standardized Precipitation Index (SPI; McKee et al., 1993) facilitates the comparison of different sources of information on precipitation in view of the mentioned challenges. It provides a common basis for comparisons between different locations, periods or seasons. The core of the SPI calculation is the fit of a distribution function to the precipitation estimates.

Previous usage of the SPI in paleoclimatology generally focussed on the index series (compare, e.g., Domínguez-Castro et al., 2008; Seftigen et al., 2013) and did not consider further information available through the transformation. We apply the SPI over moving windows of 51 years to study variations in the properties of precipitation distributions on multi-decadal time-scales. We concentrate on a regional domain where all sources of data, i.e., observations, reconstructions, and simulations are available. By applying the SPI-transformation over moving windows, we are able to evaluate and compare percentiles of the estimates as well as the moments of the distributions and the temporal changes of these distributional properties. We are essentially comparing sequences of climatologies.

Long observationally based records allow us to assess how the statistics of observed precipitation have changed over the last couple of centuries. They, in turn, provide the basis for evaluating how state-of-the art regional or global climate model

simulations and reconstructions for the Common Era (CE) compare in domains co-located with the available observations. We choose southern Great Britain as our domain of interest since there are precipitation observations available in form of the England-Wales precipitation data set (Alexander and Jones, 2000, for the period 1766 CE to present), its subdivisions, and instrumental records for Oxford (cf. Radcliffe), Pode Hole and Kew Gardens. The instrumental records start in 1767 CE, 1726

CE, and 1697 CE, respectively.

A number of precipitation reconstructions are available for the South of Great Britain. We choose the millennium-long tree-ring based data by Cooper et al. (2013) and Wilson et al. (2013) for East Anglia and Southern-Central England, respectively. We focus on an extended spring season (MAMJJ). The next section discusses our decision to concentrate on this data instead of the $\delta^{18}O$-based scaling approaches by Young et al. (2015, covering the period 1766 to present) and Rinne et al. (2013,

reconstructed values from 1613 to 1893 CE).

Regional simulations for the last 500 to 2000 years are rare. Among studies on these, Gómez-Navarro et al. (2013) describe a regional simulation with the model MM5 for Europe. Gómez-Navarro et al. (2015) compare this simulation to reconstructions for various parameters over larger regional domains within Europe. For precipitation, they compare the simulation to the gridded precipitation reconstructions of Pauling et al. (2006) for Western Europe, which is based on a set of dendroclimato-

logical and other natural proxies and documentary information. Gómez-Navarro et al. (2015) find rather good agreement in the evolution of median precipitation amounts between the reconstruction and their regional simulation for a domain including the British Isles and Ireland for the summer season. The agreement is much weaker for the spring season. They also emphasize model shortcomings and the lack of agreement in the representations of extreme climate anomalies. On the side of the reconstructions, Gómez-Navarro et al. (2015) stress the inconsistencies among the reconstructions of different parameters (i.e.,

temperature, precipitation, and sea level pressure).

Here, we compare observations and paleo-observations with each other. We additionally compare them to output from a regional simulation with the model CCLM for the European domain over the period 1645 to 1999 CE (compare Gómez-Navarro et al., 2014; Bierstedt et al., 2016). Our comparison differs from Gómez-Navarro et al. (2015) by using a different regional model, focussing on a smaller region, and by using regional time series reconstructions instead of deriving records

from gridded products. Moreover, our general focus is on precipitation, including regional instrumental series.

Our focus is not least to motivate the use of the Standardized Precipitation Index in hydroclimatic comparisons between different data sets in paleoclimatology. We use the SPI to study the consistency of the different sources of precipitation information for approximately the last 350 years. That is, we are looking how well the sources of information compare among each other. This is a limited aim, which is appropriate considering the various uncertainties especially in simulations, and

reconstructions, but also in observations. We explicitly do not expect the simulation output to agree with the instrumental and paleo-observation data on the mean precipitation amount since spatial representations differ. We also do not expect them necessarily to agree on decadal variations in precipitation because of the presence of internal variability (compare, e.g., Deser et al., 2012b, a; Swart et al., 2015) potentially masking commonly forced external signals. Thus, even a large ensemble of simulations may not necessarily represent these variations (see, e.g., Annan and Hargreaves, 2011). Since we transform precipitation to

**Table 1.** List of data sets by region, parameter, type of data, season used, and source for obtaining the data.

| Location/Region | Parameter | Type | Season | Source |
|---|---|---|---|---|
| England-Wales | Precipitation | Observations | MAMJJ | https://www.metoffice.gov.uk/hadobs/hadukp/ |
| South-West England | Precipitation | Observations | MAMJJ | https://www.metoffice.gov.uk/hadobs/hadukp/ |
| South-East England | Precipitation | Observations | MAMJJ | https://www.metoffice.gov.uk/hadobs/hadukp/ |
| Central England | Precipitation | Observations | MAMJJ | https://www.metoffice.gov.uk/hadobs/hadukp/ |
| East-Anglia | Precipitation | Reconstruction | MAMJJ | https://www.ncdc.noaa.gov/paleo-search/study/12896 |
| Southern-Central England | Precipitation | Reconstruction | MAMJJ | https://www.ncdc.noaa.gov/paleo-search/study/12907 |
| Central England | Temperature | Observations | MAMJJ | https://www.metoffice.gov.uk/hadobs/hadcet/ |
| Kew Gardens | Precipitation | Instrumental | MAMJJ | https://climexp.knmi.nl/ |
| Pode Hole | Precipitation | Instrumental | MAMJJ | https://climexp.knmi.nl/ |
| Oxford | Precipitation | Instrumental | MAMJJ | https://climexp.knmi.nl/ |
| Europe | Precipitation | CCLM Regional climate model simulation | MAMJJ | http://doi.org/10.6084/m9.figshare.5952025 |
| Europe | Temperature | CCLM Regional climate model simulation | MAMJJ | http://doi.org/10.6084/m9.figshare.5952025 |

the Standardized Precipitation Index over moving windows, our analyses essentially become comparisons between series of climatologies, thus potentially filtering shorter term internal variability.

In the following, we first introduce and discuss our choices on data sets and methodology before comparing the data sets and discussing the results. A document asset supplements this manuscript but provides only analyses that are non-essential for our conclusions.

## 2 Data

Hydroclimatic changes affect humans and the environment most on the local and regional scale. Therefore, we focus on small domains and use precipitation data. Precipitation is a more tangible variable than, e.g., drought indicators like the Palmer Drought Severity Index (PDSI). We only use the single time series records instead of gridded products to avoid the possibly spurious non-climatic variance and other stastical artifacts potentially introduced by reconstruction techniques.

We aim at describing how much agreement we can find between different sources of information for precipitation in a small domain over a period with limited instrumental data, i.e., a period when we have to rely on reconstructions from paleo-observations. Such an assessment helps to increase our confidence in the estimates by the different sources of information. In turn, it also increases our understanding of past hydroclimatic variability.

We use observationally derived data sets, reconstructions, and simulation output in our main analyses. Additionally we use further observationally derived records and instrumental station observations. Table 1 lists the sources of information. For all analyses, we use primarily the spring-summer season from March to July (MAMJJ).

Data availability motivates the choice of the regional domain. For southern Great Britain, there exist observational regional domain composite records for temperature and precipitation, precipitation reconstructions, and long instrumental records.

## 2.1 Observations

We choose the South of Great Britain as our domain of interest since there are precipitation observations available in form of the England-Wales precipitation data set (Alexander and Jones, 2000), its subdivisions, and instrumental records for Oxford (cf. Radcliffe), Pode Hole and Kew Gardens. Furthermore, there is also a long observational temperature record available for additional comparison, the Central England Temperature series (Parker et al., 1992). Croxton et al. (2006) find that the England-Wales precipitation and the Central England Temperature well represent the climate of the South of Great Britain in the late 20th century.

Alexander and Jones (2000; see also Wigley et al., 1984) describe the England-Wales precipitation (EWP) data. It is available from the Met Office Hadley Centre at monthly resolution extending back to the year 1766. The Met Office Hadley Centre also provides subdivisions of the data. We use those for South-West, South-East, and Central England. We concentrate on the England-Wales domain because there is also temperature data available in form of the long Central England Temperature series (Parker et al., 1992). Alexander and Jones (2000) describe the automated method of updating long precipitation series like the data by Wigley et al. (1984) while also ensuring the homogeneity of the data. Parker et al. (1992) similarly describe the production of temperature data to complement long-running series while maintaining quality-control and homogeneity.

The Climate Explorer (http://climexp.knmi.nl/) provides access to a number of long series of monthly instrumental precipitation observations from the Global Historical Climatology Network (Peterson and Vose, 1997). We use those from Oxford, Kew Gardens, and Pode Hole in addition to the observationally derived Met Office Hadley Centre data sets. The Climate Explorer provides monthly data for these locations from 1697 to 1999, 1726 to 1994, and 1767 to 1999 CE, respectively. The later years in the Oxford record include missing values and we therefore only use data from 1767 to 1996 CE.

## 2.2 Reconstructions

To our knowledge, there are three precipitation reconstructions for small domains from the South of Great Britain, i.e., approximately within the domain of the England-Wales precipitation and the Central England temperature. These are for East Anglia (Cooper et al., 2013), for Southern-Central England (Wilson et al., 2013), and the reconstruction for Southern England by Rinne et al. (2013). The former two use tree-ring width data for their reconstructions, the latter uses tree-ring oxygen isotopes. There is additionally the work by Young et al. (2015), who scale a $\delta^{18}O$ composite record from Great Britain to the England-Wales precipitation.

We decide only to use the two tree-ring width based records. The main reason for excluding the Rinne et al. record is that it concatenates instrumental data from Radcliffe (cf. Oxford) station for 1894 to 2003 to the reconstructed values from 1613

until 1893. This reduces the time of overlap with the England-Wales precipitation data. The reconstruction by Rinne et al. (2013) is not publicly available, but the lead author provided us with the data. We provide a short assessment of the data in a supplementary manuscript asset.

Similarly, Young et al. (2015) scale their input $\delta^{18}O$ records by precipitation and provide the input series as supplement to their paper. Our supplementary manuscript asset provides a short assessment of a scaling using this data.

In the main manuscript, we only use the data by Cooper et al. (2013) and Wilson et al. (2013) for, respectively, East Anglia and Southern-Central England in March, April, May, June, July (MAMJJ). Cooper et al. (2013) and Wilson et al. (2013) identified this extended spring as the season their tree-ring width records are sensitive to for their reconstructions of precipitation. In the following, we compare the England-Wales precipitation with the two reconstructions for the South of Britain.

Wilson et al. (2013) and Cooper et al. (2013) already discuss the limitations of their respective reconstructions. Both reconstructions represent between 30% and 35% of regional interannual precipitation variance over the 20th century. Obviously, the reconstructions suffer from the limited lengths of the available tree ring samples. This has an effect on how much low frequency variability the reconstructions can resolve. The authors note variable relationships between tree growth and environmental controls for their regions in the past. Indeed there are periods when relations between trees and precipitation are not significant. Both studies are confident in the mid- to high-frequencies of their reconstructions but emphasize that their reconstructions have weaknesses in representing extreme years when compared to the observations. Cooper et al. (2013) explicitly call their paper "preliminiary" with respect to reconstructing low frequency precipitation variability.

Young et al. (2015) find that the two reconstructions from tree-ring widths strongly differ from their own scaled $\delta^{18}O$ data. The extended spring reconstructions are basically unrelated to the $\delta^{18}O$ data. Young et al. (2015), therefore, question whether both approaches reliably represent precipitation in the South of Great Britain. After discussing possible reasons for the disagreement, Young et al. (2015) conclude that the reconstructions by Cooper et al. (2013) and Wilson et al. (2013) are valid representations of oak growth in England, but they are not reliable representations of regional precipitation variations in contrast to the $\delta^{18}O$ data of Young et al. (2015).

## 2.3 Simulations

We compare the observations and the reconstructions to output from a regional simulation with the model CCLM for the European domain over the period 1645 to 1999 as also used by Gómez-Navarro et al. (2014). Forcing for the regional simulation is from a global simulation with the MPI-ESM climate model in its COSMOS set-up (see below). We use output from 1652 onwards (Gómez-Navarro et al., 2014).

The lateral forcing of the regional simulation is output from the Millennium-simulation COSMOS-setup of the Max-Planck-Institute Earth System Model (MPI-ESM). For details, see Jungclaus et al. (2010). This version of MPI-ESM couples the atmosphere model ECHAM5, the ocean model MPI-OM, a land-surface module including vegetation (JSBACH), a module for ocean biogeochemistry (HAMOCC), and an interactive carbon cycle. For the simulation, ECHAM5 was run with a T31 horizontal resolution and with 19 vertical levels. MPI-OM used a variable resolution between 22 and 250 km on a conformal grid for this simulation. The ensemble used diverse forcings. The driving simulation for the regional simulation with CCLM

is one MPI-ESM simulation with all external forcings and a reconstruction of the solar activity based on Bard et al. (2000), i.e. with a comparatively large amplitude of solar variability.

The regional climate model CCLM simulation (Wagner, personal communication; see also Gómez-Navarro et al., 2014; Bierstedt et al., 2016) uses adjusted forcing fields relevant for paleoclimate simulations as also used with the global MPI-ESM simulation. These include orbital forcing and solar and volcanic activity. Since the regional model does not represent the stratosphere, the regional simulation considers the effect of volcanic aerosols as a reduction in solar constant equivalent to the net solar shortwave radiation at the top of the troposphere in MPI-ESM. $CO_2$ variability is prescribed and changes in greenhouse gases $CO_2$, $CH_4$, and $N_2O$ are based on data by Flückiger et al. (2002). Land-cover changes are included as external lower boundary forcing using the same data set as the MPI-ESM simulation (Pongratz et al., 2008). The presented CCLM simulation uses a rotated grid with a horizontal resolution of 0.44 by 0.44 degree and 32 vertical levels. The sponge zone of seven grid points at each domain border is removed and fields are interpolated onto a regular horizontal grid of 0.5 by 0.5 degree.

We choose the domain including grid points closest to the longitudinal and latitudinal borders 5.5W to 1.5E and 50.5 to 54.5N to represent the England and Wales precipitation domain. This selection is somewhat arbitrary but we assume it sufficiently represents the England-Wales precipitation domain to allow meaningful comparison of changes in percentiles, although not in absolute percentile values. We choose the domain 5 to 0W and 50 to 55N as simulated counterpart of the Central England Temperature. The simulated East Anglia series represents the domain 0E to 2E and 52N to 53N, and we choose the domain 2.5W to 0E and 51N to 52.5N as equivalent for Southern-Central England. All analyses are for an extended spring season from March to July since this is the seasonal focus of the reconstructions. The appendix provides a short evaluation of the simulation against the observational CRU-data (Harris et al., 2014) over the European domain. We do not apply any bias correction to the simulation output.

So far, global simulations for the last millennium have notably coarser resolutions than the 0.44 by 0.44 degree of the regional simulation we use here (compare, e.g., Fernández-Donado et al., 2013; PAGES2k-PMIP3 Group, 2015). However, compared to other regional simulations this is only a coarse resolution dynamical downscaling. Thus, one may question the benefits of the approach compared to more recent higher-resolution global simulations, e.g., with the global models CCSM4 and CESM1 (Landrum et al., 2012; Lehner et al., 2015), which have resolutions of $0.9° \times 1.25°$.

A review by Ludwig et al. (2018, including two of the present authors) emphasizes that the demand for long simulation periods limits applications of regional models in paleoclimatology to relatively coarse 50km setups. Ludwig et al. (2018) conclude that regional simulations provide more realistic distributions for precipitation in the paleo-context. Flato et al. (2013, chapter 9 of the IPCC AR5) are more ambiguous in their review but they emphasize the value of regional downscaling as a tool in addition to higher resolved global simulations.

## 3 Methods

One objective of this manuscript is to highlight how the concept of the Standardised Precipitation Index (SPI, McKee et al., 1993) adds additional perspectives on comparing various sources of information for periods with and without instrumental observations. Therefore, we shortly introduce the SPI-transformation procedure and how we use this information to subsequently compare precipitation estimates from observations, reconstructions, and a regional climate simulation.

### 3.1 The Standardized Precipitation Index – SPI

Standardising precipitation data facilitates comparing distributions between different locations, time-scales, periods, and data sources. For this purpose, McKee et al. (1993) introduced the Standardized Precipitation Index (SPI). Indeed the UK drought portal (https://eip.ceh.ac.uk/droughts) relies on the SPI, and there are recommendations to use the SPI in operational monitoring of meteorological drought (e.g. Hayes et al., 2011). Sienz et al. (2012) discuss biases of the methods.

Previous usage of the SPI in paleoclimatology generally focussed on the index series and did not consider further information available through the transformation from precipitation to SPI. For example, Domínguez-Castro et al. (2008) and Machado et al. (2011) compare SPI-series to differently derived hydroclimatic indices over approximately the last 500 years. Other studies reconstructed the SPI instead of absolute precipitation amounts (e.g. Seftigen et al., 2013; Yadav et al., 2015; Tejedor et al., 2016; Klippel et al., 2018). Lehner et al. (2012) use the SPI to compute pseudo-proxies from re-analysis data and long simulations with global climate models to test a reconstruction-method.

#### 3.1.1 Transformation

The standardized precipitation index requires fitting a distribution function to the precipitation data. There are various candidate distributions as, e.g., Sienz et al. (2012, and their references) discuss (see also Stagge et al., 2015).

In our analyses, we fit a Weibull distribution. Results differ only little if we fit Gamma or Generalised Gamma distributions (not shown). McKee et al. (1993) recommend at least 30 data points for successful distribution fits, but Guttman (1994) notes the lack of stability for small sample sizes and shows that higher order L-moments only converge for samples larger than about 60 data points.

We fit distributions over moving 51-year windows and a bootstrap procedure samples 1000 times 40 data points from each window to provide an estimate of sampling variability (presented in Appendix Figure B1). Our procedure of the SPI-calculation follows the detailed description by Sienz et al. (2012).

#### 3.1.2 Evaluation

Standardising precipitation data can avoid or at least attenuate some of the problems mentioned in the introduction. Transforming precipitation to standardised values provides further means to study the agreement or the lack thereof between different data sources.

By transforming to Standardized Precipitation Indices over moving windows, we essentially compare climatologies and potentially filter shorter term internal variability. One particular interest is to consider to which extent the different data sources describe comparable evolutions in various percentiles, e.g., representing extremes. If the transformation over moving windows filters a certain amount of internal variability, if boundary and forcing conditions are sufficiently equivalent in the simulation compared to the observed climate, and if simulated precipitation and the observed climate react equivalently to these conditions, precipitation distributions and their properties may change consistently between different sources of information. The results of Gómez-Navarro et al. (2015) give some indications that this expectation may be warranted. In the worst case, our analyses point out that one of the sources of information completely contradicts the other data sets.

For any given window, the fitted distribution parameters allow calculating various properties. For example, we can consider the changing amount of precipitation, which one would describe as average, extremely high, or extremely low for subsequent periods. In the SPI-literature, the 6.7th and 93.3th percentiles represent traditionally the regions of severe (and extreme) dryness/wetness of the probability density function. Accordingly, we subsequently compare 6.7th and 93.3th percentiles for the fitted distributions over time. Further, we can compare the moments of the distributions. We choose to show the square-root of the Weibull distribution variance, i.e., the Weibull standard deviation over sliding windows. The Appendix C shows parameters for the distribution fits.

The fitted parameters allow further analyses, e.g., we can compare how likely a reference amount of precipitation is for different periods. We do this for 50th, 6.7th, and 93.3th percentiles in a reference year. We choose 1815 CE as reference year, since it is included in all data sets and it allows potentially equivalent analyses of the PMIP3 past1000 simulations (e.g., Schmidt et al., 2011).

Agreement on changes in percentiles and standard deviation increases our confidence in our understanding of forced and unforced changes in precipitation variability and projected future precipitation variations. Disagreement on estimated changes may highlight differing internal climate variability between observed, reconstructed, and simulated data or it may signal that the simulation does not correctly capture forced variations.

## 3.2 Smoothing

Performing the transformation to standardised precipitation over 51-year windows results in smoothed estimates. For convenience, we additionally plot smoothed time series in a number of Figures. Filtered series are solely used for visualisation.

We use a Hamming window. In most cases, this has a length of 51 points but we also occasionally use different window lengths. The 51-point Hamming filter represents a different frequency cut-off than a simple 51-year moving median or moving mean as can be obtained from fitting the distributions over 51-year moving windows.

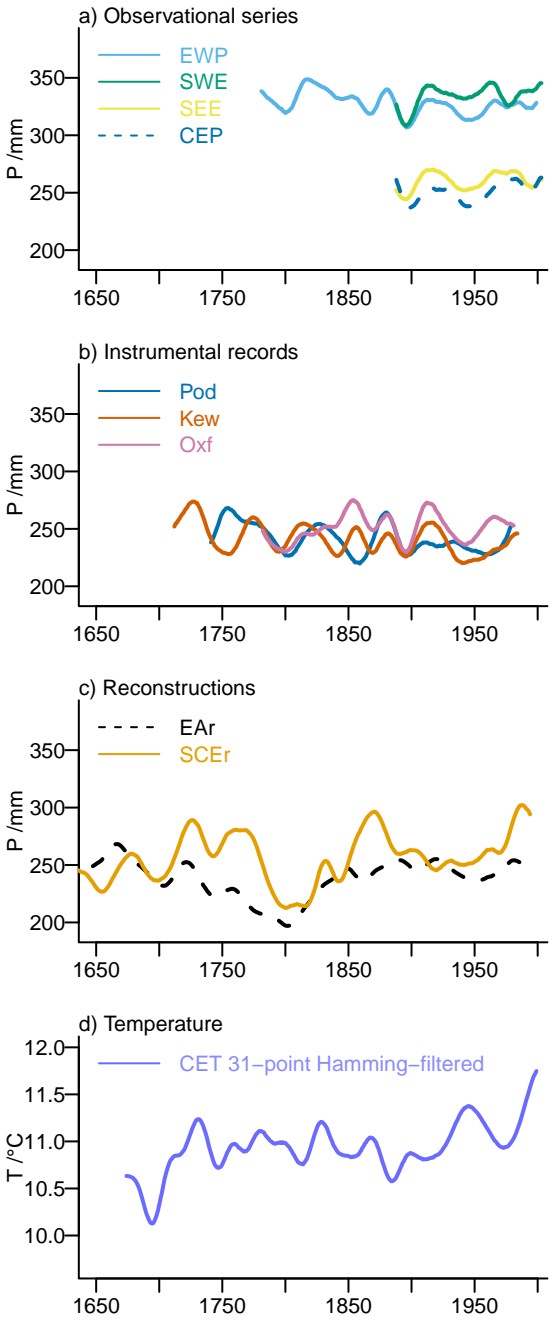

**Figure 1.** Visualisation of the observation based records for the extended spring season March to July (MAMJJ). We show 31-point Hamming-filtered time series for a) the Met Office Hadley Centre observational precipitation series for England-Wales (EWP), South-West (SWE), South-East (SEE), and Central England (CEP), b) the instrumental precipitation series for Pode Hole (Pod), Kew Gardens (Kew), and Oxford (Oxf), c) the precipitation reconstructions for East Anglia (EAr) and Southern-Central England (SCEr), and d) the Central England Temperature (CET) data.

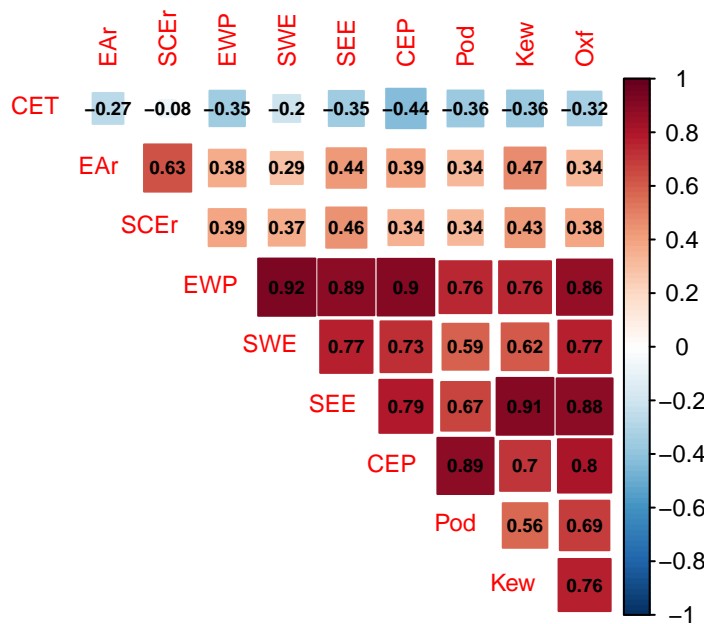

**Figure 2.** Correlation matrix for complete correlations between the observation or paleo-observation based data sets Central England Temperature (CET), East Anglia precipitation reconstruction (EAr), Southern-Central England precipitation reconstruction (SCEr), England-Wales precipitation (EWP), South-West England precipitation (SWE), South-East England precipitation (SEE), Central England precipiation (CEP), Pode Hole precipitation (Pod), Kew Gardens precipitation (Kew), and Oxford precipitation (Oxf). Complete correlations mean, we only use the years 1873 to 1994 for which all records have data. The season for all records is MAMJJ.

## 4 Results

### 4.1 Relations among data sets

#### 4.1.1 Observational data and reconstructions

Figure 1 provides a first impression of the observational and reconstruction data we use in the following. All series are for
5   the extended spring season from March to July on which we focus. Panels show 31-point Hamming-filtered time series. These allow a better qualitative assessment of the commonalities between the data sets and the differences compared to, e.g., 11-point or 51-point Hamming-filtered time series. Observational precipitation series from the Met Office Hadley Centre for South-West, South-East, Central England, and England-Wales show high agreement in their variations on these time-scales (see Figure 1a). The instrumental time series for Kew Gardens and Pode Hole show more disagreement in certain periods for the considered
10   smoothing, i.e., they even evolve oppositely at certain times (see Figure 1b). The instrumental data for Oxford appears to agree better with the data for Kew Gardens, which is to be expected from the geographic locations of the stations. Visually, both reconstructions agree less well with the observational series and with each other than the observational data does (see Figure 1c). Figure 1d adds the Central England temperature data for MAMJJ for completeness sake.

Correlation matrices (Figure 2, and supplementary manuscript asset) and scatterplots (see manuscript asset) emphasize the differing agreement between the various data sources even more clearly on interannual time-scales. Figure 2 presents the correlation matrix for complete observations, i.e. for the period 1873 to 1994 when all records have data. Correlation coefficients change slightly if we consider pairwise complete records. Relations among precipitation data sets are always positive. They are very strong between the England-Wales data and its subdivions, between the Kew Gardens series and the South-East England data, between the Pode Hole series and the Central England data, and between the Oxford record and the South-East England data as well as the England-Wales precipitation. The relation between the two reconstructions is also rather strong over the sub-period. Correlations are, however, weaker between the reconstructions and the observed series.

There is a generally negative relation between the Central England temperature and the precipitation data sets. It is weakest for the Southern-Central England reconstruction but also rather weak for the East-Anglia reconstruction and the South-West England record from the Met Office Hadley Centre. Scatterplots emphasize that even the temperature-precipitation relations with larger correlations scatter widely (not shown). Temperature-relations are stronger for the observationally based data from the Met Office Hadley Centre and the instrumental series for the summer season June to August (not shown).

Correlations for non-overlapping 11-year averages are positive and strongest between the England-Wales precipitation and the two instrumental series (not shown, see supplementary manuscript asset, calculated for the period 1767 to 1986). This analysis gives also reasonable correlations ($r \approx 0.51$) between the pair of reconstructions and between the instrumental series. Otherwise, correlations for this resolution are weak. Correlations with the Central England temperature data are largest for the non-overlapping 11-year averages of the Kew Gardens instrumental series.

### 4.1.2 (Paleo-)observational data and regional simulation output

Figure 3 presents the two reconstructions and the England-Wales precipitation in comparison to the respective data from the regional simulation. All data are again for the extended spring season from March to July (MAMJJ), and the panels zoom in on the period of the regional simulation. We show the interannual time series and the 51-point Hamming-filtered representation.

Considering the evolution of the records, the 51-point Hamming-filtered time series show pronounced differences besides some common features for the reconstructions for Southern-Central England (Wilson et al., 2013) and East Anglia (Cooper et al., 2013) (black lines in Figure 3a and b) similar to the representations in Figure 1. Both reconstructions feature a relative precipitation minimum centered on approximately the year 1800. The Southern-Central England reconstruction additionally displays a relative minimum in the early 20th century.

The observed England-Wales precipitation is available at monthly resolution from the year 1766 onward. The Hamming-filtered time series shows markedly less multi-decadal to centennial variability compared to the reconstructions, but the observations have much more interannual variability than the reconstruction for East Anglia and slightly more variability than the reconstruction for Southern-Central England (Figure 3c, black line). The filtered England-Wales time series also displays a slightly negative trend.

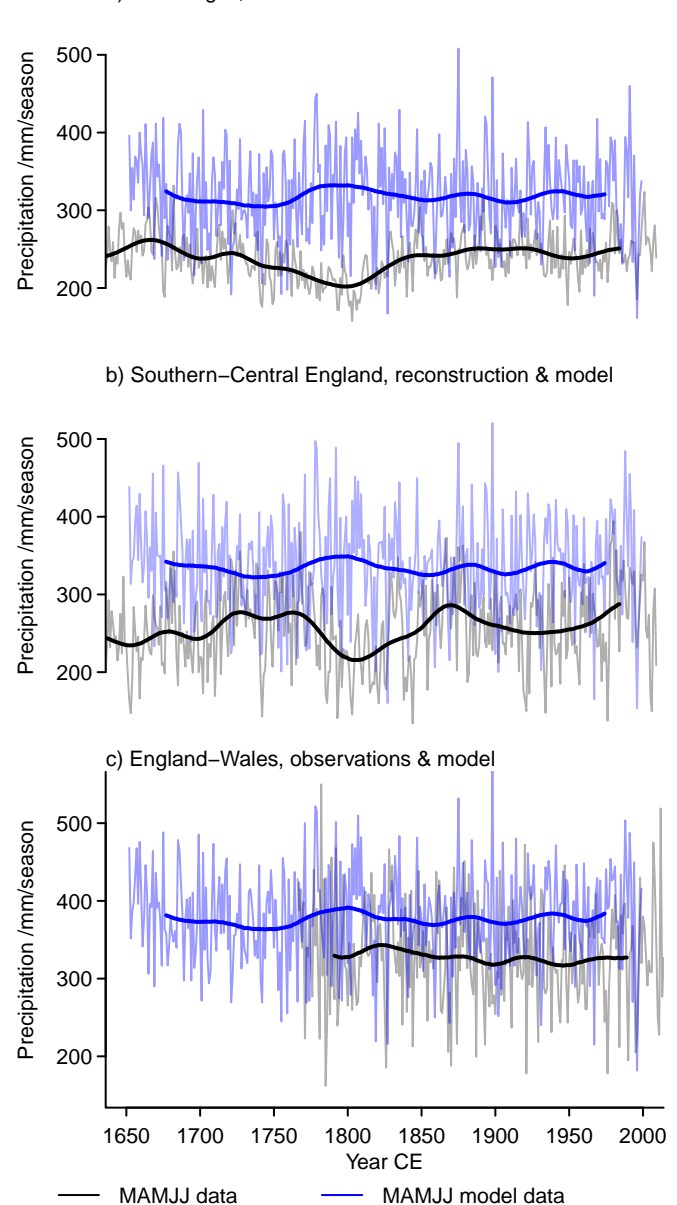

**Figure 3.** Extended spring (MAMJJ) precipitation in (paleo-)observation based data and simulation output, a) East Anglia precipitation in reconstruction (black) and regional model (blue), b) Southern-Central England precipitation in reconstructions (black) and regional simulation (blue), and c) England-Wales precipitation in observational data (black) and regional simulation (blue). We show interannual data (light colors) and 51-point Hamming-filtered data (solid colored).

Differences between the simulated regional records are generally smaller (blue lines in Figure 3). Existing differences highlight the spatial heterogeneity of precipitation. A general feature for all regions is that excursions of the filtered simulation output often, but not always, are opposite to those of the reconstructions or observation time series.

There is an obvious bias in the absolute amounts between the simulation output and the other data sets. The simulation output series give larger precipitation amounts. We do not try to attribute this difference. We note that it is not as prominent for the more local comparison with the data from Rinne et al. (2013) for May to August and the bias is generally slightly negative for the summer season June to August for England-Wales precipitation (not shown, see supplementary manuscript asset). We assume that the differing spatial representations sufficiently explain the mismatch. However, the change of sign in the bias for the summer season suggests that the simulation overestimates spring precipitation, underestimates summer precipitation, and the positive spring bias is larger than the negative summer bias. See also Appendix A for a comparison of the simulation to observational data over the full European model domain.

This initial comparison already shows varying levels of agreement for the chosen data sets derived from observations and the reconstructions. It highlights that the relation between the reconstructions and the observational data sets are weaker than between the instrumental data and the observational indices on interannual time-scales. Note that the regional observational indices include information from the instrumental data. On longer time-scales the reconstructions align less well among each other than the observationally derived time series. However though possibly not surprisingly, the local, purely instrumental series also show more disagreement among each other than the derived larger domain products. Filtered regional time series evolve often visually oppositely in the simulation compared to the reconstructions and the observations.

So far, we used the precipitation and temperature data. In the following, we mainly use the information obtained via the transformation to standardised precipitation indices.

## 4.2 Comparing standardised precipitation data

Figure 4 to 6 add, respectively, the comparisons of the wet, i.e. 93.3th, percentile, the dry, i.e. 6.7th, percentile, and the square root of the Weibull distribution variance to the comparison of the interannual and filtered time series in the previous section.

### 4.2.1 Observations vs. Reconstructions

Since they represent different regions, we do not expect agreement in the absolute precipitation amounts representing wet conditions between the England-Wales precipitation data and the reconstructions in Figure 4a. We note that the difference between the wet percentile for the England-Wales precipitation and the reconstructions is larger than for the average amounts, indicating a wider distribution for the data based on instrumental observations. Precipitation histograms confirm this (not shown). On the other hand, differences are smaller for the dry percentile (Figure 5). Nevertheless, this is a sign that the reconstructions underestimate the width of the precipitation distributions of 51-year window climatologies.

Reconstructed and observation-based time series show a slightly opposite trend for the wet percentile over much of the period of the observational England-Wales time series (Figure 4). Smaller scale variations in the beginning of the wet percentile series

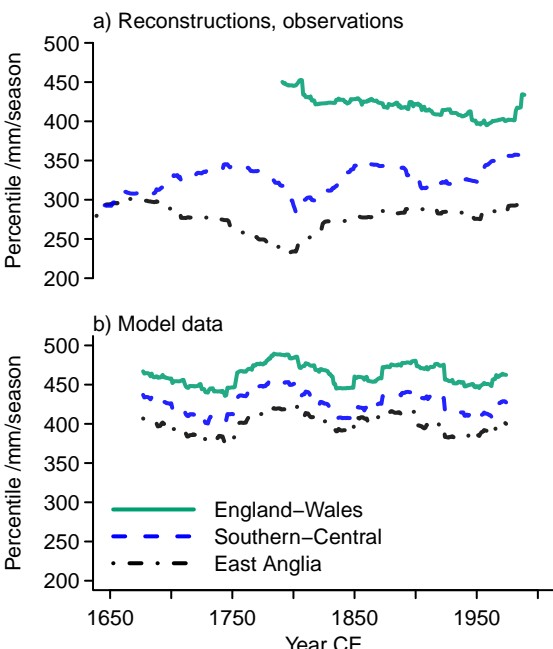

**Figure 4.** Visualisation of the MAMJJ precipitation amount identified as severely wet (93.3th percentile) over 51-year windows for England-Wales (green solid lines), Southern-Central England (blue dashed lines), and East Anglia (black dash-dotted lines) in a) reconstructions and observations, and b) simulations.

are also opposite. The dry percentile series lack the clear overall trend but multidecadal variations evolve oppositely between reconstructed and observed dry percentiles (Figure 5).

The opposite trends in the wet percentiles mean that the observed 93.3th, i.e. wet, percentile represents lower precipitation amounts in the middle of the 20th century compared to the late 18th century, while the reconstructed wet percentile represents larger precipitation amounts in the middle of the 20th century compared to the late 18th century (Figure 4). Similarly the opposite multidecadal variability in the 6.7th, i.e. dry, percentiles of reconstructions and observations means that when the reconstructions represent a drying of the dry percentiles, the observations indicate that very dry conditions are already identifiable for larger precipitation amounts in a period and vice versa (Figure 5). Generally, the series for the severe to extreme dryness and wetness percentiles reflect the smoothed evolution of the respective data set.

We note that the data of Rinne et al. (2013) for Southern England in summer display an apparent opposite evolution of wet percentiles for the period of overlap between reconstruction and observations. On the other hand dry percentiles agree well (not shown, see supplementary manuscript asset).

Parameters for the fitted distributions also allow to evaluate the moments of the distributions. Estimates for the Weibull standard deviations (SD, Figure 6) differ between observations and reconstructions as expected from the previously noted differences in percentiles. The reconstructions do not show a clear evolution in the Weibull standard deviations. The observations show a slight reduction in the standard deviation until the middle of the 20th century, with a strong increase afterwards.

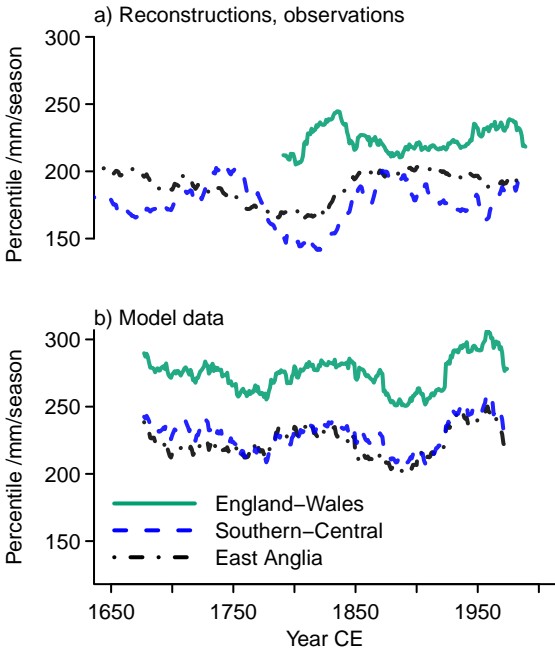

**Figure 5.** Visualisation of the MAMJJ precipitation amount identified as severely dry (6.7th percentile) over 51-year windows for England-Wales (green solid lines), Southern-Central England (blue dashed lines), and East Anglia (black dash-dotted lines) in a) reconstructions and observations, and b) simulations.

### 4.2.2 Simulation output

The simulated time series in Figure 3 show large similarities between regions. This is also the case for the wet and dry percentiles as well as for the standard deviations. Indeed, the respective statistics evolve simultaneously among the different regions, and the standard deviations overlap (Figures 4 to 6).

5     Thus, differences between regional domains are smaller for their simulated representations compared to the observed or reconstructed records. They are slightly more notable for the moving window statistics compared to the Hamming-filtered series. Dry percentiles are very similar for East Anglia and for Southern-Central England in the simulation but wet conditions require larger precipitation amounts for Southern-Central compared to East Anglia. Appendix B highlights that this may be due to sampling variability. Smoothed simulated data and wetness percentiles evolve similarly, but opposite evolutions of the

10   dryness and wetness percentiles result in widening and shrinking of the distributions after approximately the year 1800.

### 4.2.3 Simulation output vs. observationally derived data and reconstructions

Simulations and reconstructions agree only minimally (Figures 4 to 6). The simulation appears to agree sligthly with the reconstruction for Southern-Central England in the late 19th century in the wet percentile (Figure 4). However, then the dryness percentile evolve in an opposite way (Figure 5).

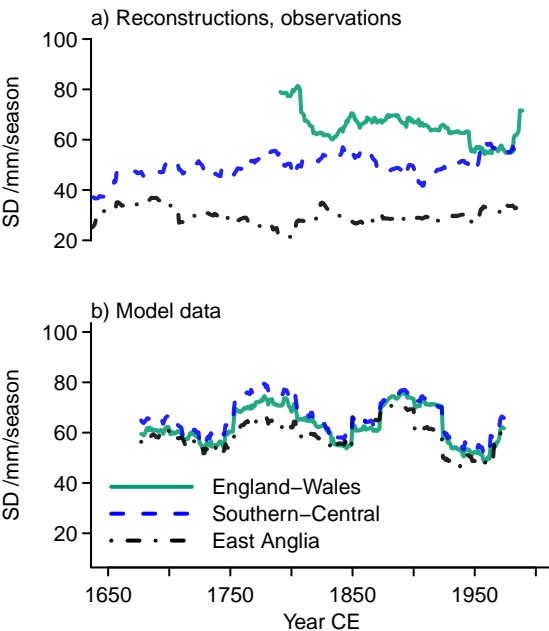

**Figure 6.** Visualisation of Weibull standard deviations over 51-year windows for MAMJJ precipitation for England-Wales (green solid lines), Southern-Central England (blue dashed lines), and East Anglia (black dash-dotted lines) in a) reconstructions and observations, and b) simulations.

This apparent opposite evolution is the most common feature when comparing the percentiles derived from the simulation and from the reconstructions. When the percentile series for the reconstructions show minima, the simulation commonly shows maxima and vice versa. Obviously, using an ensemble of regional simulations probably would show different trajectories. This does not preclude per-se that the model is capturing basic physical characteristics of precipitation variability in northwestern
Europe.

The smoothed representations of the simulation output and the smoothed observed England-Wales precipitation show only small multidecadal variations, which appear to be more or less opposite in simulated and observed estimates (compare above Figure 3). The wet percentiles do not show any agreement although they both have a relative maximum in the late 18th century (Figure 4). On the other hand, the dry percentiles show approximate agreement in their evolutions with maxima in the early 19th
century and in the middle of the 20th century (Figure 5). Similarly the Weibull standard deviations show some commonalities between the simulated representation of the England-Wales precipitation and the observations (Figure 6).

We note that there is neither any clear commonality nor any overly opposite evolution in the dry percentiles when comparing the regional simulation to the reconstruction for Southern England summer precipitation by Rinne et al. (2013, not shown, see supplementary manuscript asset). The wet percentiles, however, evolve oppositely in the 18th century but then show a common
positive trend in the 19th century (not shown, see manuscript asset).

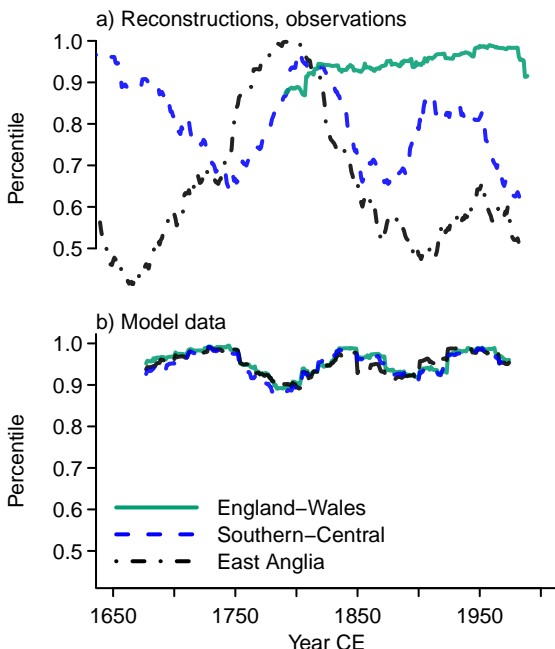

**Figure 7.** Visualisation of which percentiles the 93.3th percentile MAMJJ precipitation amount for the reference window centered in year 1815 CE represents over time for England-Wales (green solid lines), Southern-Central England (blue dashed lines), and East Anglia (black dash-dotted lines) in a) reconstructions and observations, and b) simulations.

## 4.3 Changes in probability of certain precipitation amounts

In the methods section, we describe the procedure of calculating standardized precipitation indices over moving time windows. We obtain a distribution fit for each time window. The parameters of the fit for a window allow us to identify the probability of a precipitation amount for the respective window. Figures 7 to 9 present changes in the probability of certain amounts of precipitation, i.e. lines are the changing percentiles represented by a given amount of precipitation over time. The Figures show these changes for the precipitation amounts representing 93.3th, 50th, and 6.7th percentiles, respectively, in a reference window. For this comparison, the reference is the distribution of precipitation in the window centered around the year 1815 CE. We estimate and plot the percentiles that correspond to these reference precipitation amounts in other time windows.

The England-Wales precipitation shows a slight increase over time in the reference 93.3th percentile in the year 1815 CE (Figure 7a). Recently, there is a steep decrease in the series. Similarly, the 50th percentile for 1815 CE represents slightly larger percentiles over time (Figure 8a). On the other hand, there are weak multi-decadal variations in the series for the 6.7th percentile in the observations, and the 6.7th percentile from 1815 CE may become slightly less likely over time (Figure 9a).

Before turning to the reconstructions, we shortly note that the simulations show similar trajectories for all three percentile values and all three regions. There are not any obvious trends, but the series show multidecadal variations. The window centered in the year 1815 CE falls within a minimum or at the end of a minimum. The respective precipitation amount generally

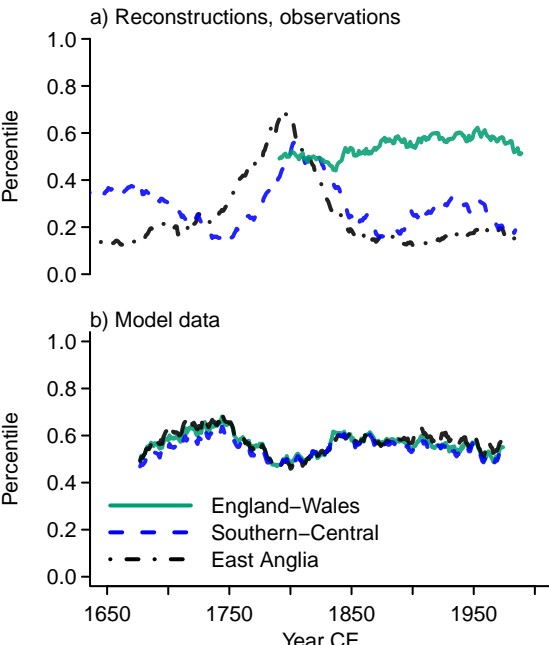

**Figure 8.** Visualisation of which percentiles the 50th percentile MAMJJ precipitation amount for the reference window centered in year 1815 CE represents over time for England-Wales (green solid lines), Southern-Central England (blue dashed lines), and East Anglia (black dash-dotted lines) in a) reconstructions and observations, and b) simulations.

represents larger percentiles before the time window centered in 1815 CE. After this time window, the 6.7th and 93.3th percentiles both approach a maximum in the series (Figures 7b and 9b). However, the 93.3th percentiles reach it about the year 1850 CE and the 6.7th percentile only in approximately the year 1900 CE, when the 93.3th percentile is again in a relative minimum. Thus, the wet and dry percentiles evolve oppositely from the early 19th century onwards, i.e. the distribution widens

and shrinks since approximately the year 1850 CE. The median reference for 1815 CE also represents larger percentiles later but there is a slight decreasing trend from approximately the mid-19th century to the end of the simulation (Figure 8b).

The reconstructions for East Anglia and Southern-Central England have some peculiar features (Figures 7a to 9a). For one, it is not ideal to choose a reference year from the period around 1800 CE. The 6.7th percentile in 1815 CE is much less likely earlier and later in both regions (Figure 9a). Similarly, average precipitation around 1815 CE represents approximately

10 the 20th percentiles in earlier and later periods for East Anglia (Figure 8a) but also represents much smaller percentiles in later periods for Southern-Central England. Severe and extreme wet conditions from this period may even represent long-term average conditions for East Anglia (Figure 7a). We note that comparisons to the data by Rinne et al. (2013) do not feature such peculiarities (not shown) but using a simple scaling approach for the $\delta^{18}O$ data of Young et al. (2015) gives similar results prior to approximately the year 1850 CE (not shown, but compare information given in the supplementary manuscript asset).

In general, there are not any clear common evolutions between the different data sets before the 20th century. Only the dry percentiles in the simulation and the observations may evolve similarly (Figure 9). Interestingly, there is an apparent contrast

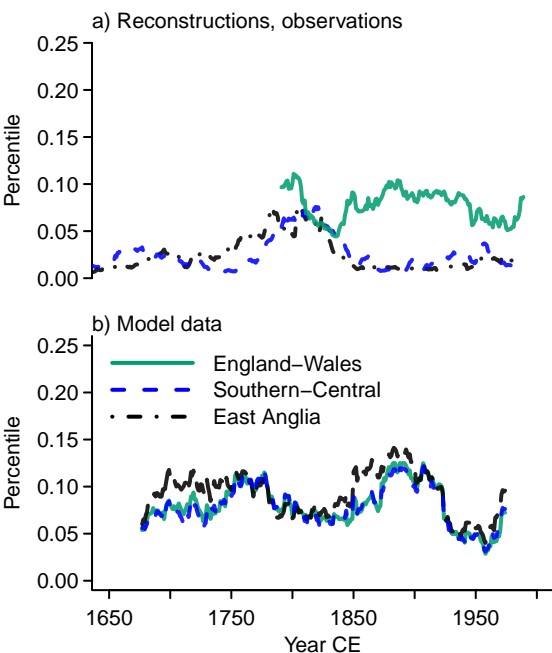

**Figure 9.** Visualisation of which percentiles the 6.7th percentile MAMJJ precipitation amount for the reference window centered in year 1815 CE represents over time for England-Wales (green solid lines), Southern-Central England (blue dashed lines), and East Anglia (black dash-dotted lines) in a) reconstructions and observations, and b) simulations.

between simulation and reconstructions with potentially opposite evolutions prior to the 20th century in all shown series. In the 20th century, on the other hand, some commonalities may be inferred at least for the series representing the reference 93.3th percentile (Figure 7).

  Most prominent in these analyses is that the distributions for reconstructed precipitation show large shifts to larger precip-
itation amounts compared to the simulations and observations. In contrast, the simulation and observations vary only within a rather narrow range. This may relate to the weaknesses of the reconstructions in representing not only low-frequencies but also extremes (compare Cooper et al., 2013; Wilson et al., 2013). The regional simulation and the reconstructions show again an apparent opposite evolution for East Anglia and Southern-Central England. All sources of information tend to show shifts in the probability of precipitation amounts.

**4.4  Relation between Temperature and Precipitation**

The high amount of internal variability on local and regional scales complicates the comparison among different data sources when studying small regions. We only shortly explore the interrelation between the regional temperature and the precipitation variability. We show how interannual correlations between the precipitation records and temperature series evolve over time. Figure 10 plots sliding interannual correlations for 51-year windows between the observed and reconstructed precipitation data
and the Central England temperature as well as the correlation between simulated England-Wales precipitation and simulated

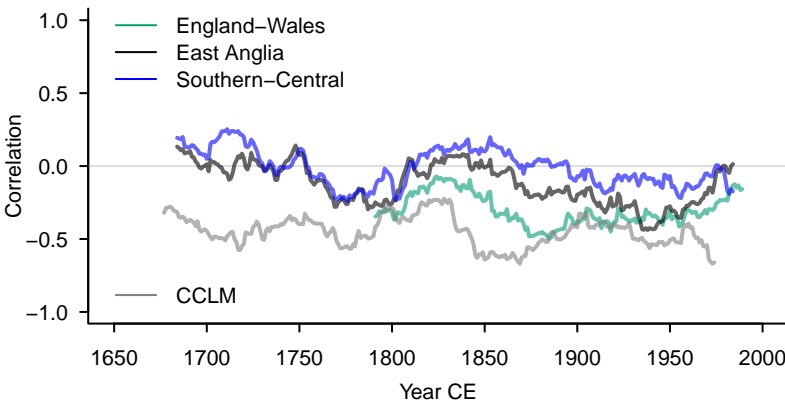

**Figure 10.** Interannual correlations over 51-year windows between extended spring (MAMJJ) Central England Temperature and various precipitation records: extended spring (MAMJJ) precipitation series for observational England-Wales-precipitation (green), reconstructed East Anglia precipitation (black), reconstructed Southern-Central England precipitation (blue). The grey line is for the simulated representations of the England-Wales MAMJJ precipitation and the Central England Temperature in MAMJJ.

Central England temperature. We plot correlations for the untransformed precipitation records. All records are for the MAMJJ-season.

We expect variability of moving correlation coefficients simply due to sampling variability (Gershunov et al., 2001). For example, a bootstrap procedure following Gershunov et al. (2001) suggests a 90% credible interval for 51-year moving win-
dow correlations of between approximately $-0.59$ and approximately $-0.21$ for a correlation of approximately $-0.43$ between simulated Central England Temperature and England-Wales precipitation over the full period. That is, variations in Figure 10 are probably within the sampling variability estimates for 51-year moving window correlations. That further implies that for overall uncorrelated data we can expect some windows to show statistically significant correlations. We do not show significance levels in Figure 10. We note that for 51-year windows and the time series characteristics of the data (e.g., approximately
uncorrelated noise for seasonal precipitation), one may regard absolute values of correlation coefficients larger than $0.23$ as statistically significant at the 5% level.

On interannual timescales and over 51-year moving windows, all data sets evolve similarly in Figure 10. However, observed and reconstructed data show weaker correlations in the late 20th century, while the correlation strength increases in the regional simulation. Both reconstructions do not show any statistically significant relation between temperature and precipitation over
the full period. The reconstruction for East Anglia is intermittently negatively correlated with the temperature data. The observations show a notable negative relation from the second half of the 19th to the mid-20th century. Only correlations between the regional simulation temperature and precipitation are negative and relatively strong ($r \approx 0.5$) throughout the full period.

The observed negative relation is well known. For example, Crhová and Holtanová (2018) show a slightly negative correlation between temperature and precipitation in observations over the southern British Isles in spring and summer. They also
show that regional climate simulations usually capture this feature successfully.

## 5  Discussions

Our understanding of hydroclimatic changes for future and past climates increased notably in recent years (compare, e.g., PAGES Hydro2k Consortium, 2017), especially for drought (see Cook et al., 2018). Nevertheless comparing our various sources of information for past hydroclimatic changes remains challenging (compare PAGES Hydro2k Consortium, 2017).

Hydroclimate comparisons between different data sources may focus on floods, on drought using indices like the PDSI, or on precipitation, including the SPI used in this study. Humans notice effects of climatic changes mostly on local to regional scales. We expect that changes in precipitation are of larger immediate relevance for local communities than changes in drought indices. Thus, we study precipitation changes in regional small domains.

In this section, we more extensively discuss the SPI, our data choices, and our results. We also discuss additional data sets
presented in the supplementary manuscript asset.

### 5.1  Method

Much research on hydroclimatic variability focusses on drought because of its effects on society and environment. Based on criteria suggested by Keyantash et al. (2002), the Interregional Workshop on Indices and Early Warning Systems for Drought proposed the Standardized Precipitation Index (SPI, McKee et al., 1993) as common index to monitor meteorological drought
(Hayes et al., 2011). The SPI should complement previously used indices and facilitate comparability between different regions. Raible et al. (2017) find the SPI to be a reliable drought index for Western Europe including the British Isles. The standardisation allows further applications, e.g., flood monitoring (Seiler et al., 2002), and the easy comparison of normal, dry, and wet conditions between different sources of data.

The SPI uses only precipitation, which makes it an ideal and relatively straightforward tool for comparing hydroclimatic data
between different data sources. Precipitation is a standard output of simulations, long instrumental records exist for various locations, and a number of reconstructions exist as well although paleo-observations may represent soil moisture rather than precipitation.

However, as the SPI uses only precipitation, it is of less value when the interest is in, e.g., the water supply, runoff, or streamflow (but see Seiler et al., 2002). The focus on precipitation also limits the applicability for studying temperature sensitive
parts of the hydrological cycle and impacts on biological and anthropogenic systems (e.g., PAGES Hydro2k Consortium, 2017; Keyantash et al., 2002; Hayes et al., 2011; Van Loon, 2015).

Most importantly, the interpretation of our results relies on the robustness of the SPI-transformations. Sienz et al. (2012) highlighted that the Weibull distribution performed better in transforming the England-Wales precipitation data on a monthly time-scale compared to a number of other distributions. However, other distributions outperformed the Weibull distribution for
other data sets and other SPI time-scales.

We fit distributions over sliding 51-year windows. While we thus use more data points than recommended by McKee et al. (1993), we still use less than the 60 points for which Guttman (1994) finds convergence of higher order L-moments. Appendix Figure B1 shows 95% intervals of a bootstrap procedure sampling 1000 times 40 data points from each window and fitting

distributions to these samples. Uncertainty on the fitted distributions varies in size over time and between data sets. Indeed, there are periods when sampling variability is so large that apparent differences in distributional properties between periods are not significant for most sources of information.

In a sense, the SPI calculations provides us with information on the climatological precipitation distributions over moving windows. The comparison becomes therefore an assessment of the changes in the climatology between different 51-year periods. This climatology does not only provide information on the mean state but also further derived statistics, the extreme percentiles for the individual windows, and the variability in these periods.

If this climatology for the observations is the target climatology, an ensemble of climate simulations should sample this distribution interannually following the paradigm of a statistically indistiguishable ensemble (Annan and Hargreaves, 2011). Our analyses compare how well the climatologies agree in different sources of data.

## 5.2 Data

### 5.2.1 Observations

Starting from the available regional climate simulation, we choose the region for our study based on the availability of precipitation observation and reconstruction data. There are long records of instrumental measurements of climate parameters for a number of locations in Europe. The British Isles are unique because there exist long observation based indices for precipitation and temperature in form of the England-Wales precipitation data (Alexander and Jones, 2000) and the Central England Temperature data (Parker et al., 1992) besides the long instrumental precipitation observations, e.g. in southern Great Britain, for Kew Gardens, Oxford, and Pode Hole. Additionally there are instrumental records from Inverness, Edinburgh, and Manchester starting in the 1780s, which we do not use because of their northern locations. For Ireland, Murphy et al. (2018) provide a monthly rainfall series starting in the year 1711, which we do not discuss here either because of its distance to our study region.

The Central England and England-Wales observation indices are good representations of the late 20th century climate of the South of Great Britain according to Croxton et al. (2006). Our Figure 2 also shows the strong correlation between the various precipitation records based on observations. Note that the composite series naturally rely on the instrumental series. Weakest relations occur for the instrumental series of Pode Hole with the sub-regional series for South-West England and for the relation between the two instrumental records from Pode Hole and Kew Gardens. Frank et al. (2007) noted the uncertainties in early instrumental temperature observations. Additionally, the very early data in the Central England Temperature data includes non-instrumental indirect data to infer past temperature. Similarly, early precipitation observations require rigorous quality control. In this context, Woodley (1996) reviews the history of England and Wales as well as Scotland precipitation data.

Figure 2 further shows the negative relation between temperature and precipitation for our domain of interest. Tout (1987) does not find any changes in the negative relation between England-Wales precipitation and Central England Temperature for the summer season from June to August between 1766 and 1980.

### 5.2.2 Reconstructions

Paleo-reconstructions of the recent past have made notable progress both in the spatial coverage and in the quality of the reconstructions by incorporating so far unexplored data sources and new methods. Küttel et al. (2010), for example, highlight the importance of ship-based observations recorded in log books for reconstructing large-scale fields. Initiatives like oldweather.org or ACRE (Atmospheric Circulation Reconstructions over the Earth, www.met-acre.org) are invaluable for such efforts and also aid reanalysis projects like the twentieth century reanalysis (Compo et al., 2011), the reanalysis of global fields for the period 1600 to 2005 by Franke et al. (2017), or the last millennium climate reanalysis (Hakim et al., 2016).

For the hydroclimate, there are a number of gridded reconstructions covering the European domain. Continental domain gridded precipitation reconstructions are, e.g., Pauling et al. (2006), Casty et al. (2007), and Franke et al. (2017). Reconstructions of drought indices like the Palmer Drought Severity Index (PDSI) exist as gridded products, too, for various regions of the world including Europe (The Old World Drought Atlas, Cook et al., 2015). These products allow assessing the quality of the hydroclimate in paleoclimate simulations (Smerdon et al., 2015).

The availability of observational data and regional reconstructions motivates our domain choice covering southern Great Britain. We decide to use regional precipitation reconstructions instead of gridded products to minimise the effect of the reconstruction method on the results. We focus on precipitation as it allows the direct comparison with long instrumental records and it is a parameter directly experienced by people.

We mainly focus on two reconstructions based on tree-ring widths measurements (Cooper et al., 2013; Wilson et al., 2013). These authors calibrate their tree-ring data against gridded precipitation beyond their target regions of Southern-Central England and East Anglia, respectively. Thereby the reconstructions are possibly biased beyond their respective regions of interest. They compare their reconstructions against the long instrumental records and find a lack of stability of the relation to the instrumental data. They discuss the limitations of their reconstructions representing less than 40% of the regional precipitation variance over the 20th century.

Although the reconstructions show a notable amount of low frequency variability, Cooper et al. (2013) cautions against too much confidence in the low frequency precipitation variability in their reconstruction. Wilson et al. (2013) and Cooper et al. (2013) emphasize the weaknesses of their reconstructions in representing extreme years. On the other hand, both are confident in the mid- to high-frequencies of their reconstructions.

Both, Wilson et al. (2013) and Cooper et al. (2013) discuss the possibility that the tree-species used for their reconstructions were less sensitive to precipitation over certain periods, e.g., the early 19th century. That is, the proxies, theoretically representing a precipitation signal, also contain a temperature signal, for instance, if they are sensitive to soil moisture. Wilson et al. further suggest an effect of the Industrial Revolution and the associated pollution on the trees in their selection. Wilson et al. (2013) also discuss the reliability of the instrumental data but conclude this is likely not an issue.

Besides these two tree-ring width based reconstructions, the works by Rinne et al. (2013) and Young et al. (2015) use $\delta^{18}O$ data to reconstruct precipitation for Southern England and Great Britain respectively. We shortly discuss results for both reconstructions below. Rinne et al. (2013) calibrate and scale their local isotope data from 1613 to 1893 CE against the station

observations from Oxford for the period 1815 to 1893 CE and concatenate the reconstruction with the observations for 1894 to 2003 CE. They target an extended summer season from May to August. Young et al. (2015) use the England-Wales summer, June to August, precipitation as scaling target for a composite of eight isotope records from Scotland, Wales, and England for the period 1766 to 2012 CE. Both publications by Rinne et al. (2013) and Young et al. (2015) note the differences of their reconstructions to the tree-ring width based works by Wilson et al. (2013) and Cooper et al. (2013).

Young et al. (2015) conclude that these differences make it unlikely that the tree-ring based works and their $\delta^{18}O$ based work represent the same environmental parameter, and they emphasize the lack of a calibration against regional precipitation data. They further discuss the reasons given by Wilson et al. (2013) for the lacking stability of the Wilson et al. reconstruction, namely, different climate-sensitivity of the trees, unreliable instrumental data, and pollution. Young et al. (2015) conclude that their own data reliably reflects precipitation while the tree-ring widths most likely represent the combination of various environmental influences on tree growth instead of a single climate parameter.

### 5.2.3 Simulations

In contrast to present-day and future scenario regional simulations, the 0.44 by 0.44 degree resolution of our CCLM simulation represents a comparatively coarse resolution dynamical downscaling. Sørland et al. (2018) discuss how the use of a model-chain including global and regional climate simulations assists studies on regional climates. Besides other models, they also use CCLM in a 50km setup comparable to the simulation used here. Their work emphasizes that improved representation of regional climate in a regional simulation is not solely due to the increased resolution but may be due to different strategies in model-building and tuning. Pinto et al. (2018) study global and regional simulations, including CCLM-simulations, for southern Africa. They explain differences in results from regional and global simulations by an interplay between the representation of sub-grid-scale processes in the different models and factors related to the increased resolution. Blenkinsop and Fowler (2007) note that regional climate models may be deficient in their ability to model persistent low precipitation episodes for the British Isles, which has repercussions for their representation of drought events.

The review by Ludwig et al. (2018) reports more realistic distributions for precipitation in regional paleoclimate simulations. Flato et al. (2013, chapter 9 of the IPCC AR5) review the progress of regional downscaling and high-resolution modelling. They emphasize that the skill of such exercises depends on the model used, the season, the domain of interest, and the considered meteorological variable. They highlight studies showing that there is not a linear increase in simulation skill towards higher resolutions. Higher resolutions typically provide more reliable estimates of extremes, including heavy rainfall. Flato et al. (2013) view regional modelling as a valuable extension of global climate modelling.

The quality of the simulated precipitation still strongly depends on the parameterisations implemented in the regional climate model. Precipitation, especially convective precipitation events, are still sub-grid processes, even within regional climate models. Concentrating on accumulated amounts on seasonal time-scales and their long-term changes, however, allows at least a more robust comparison of simulated precipitation to observed and reconstructed data.

Shortcomings of the various data sets limit our expectations to what extent they can reflect comparable variations among each other.

### 5.3 Discussion of Results

#### 5.3.1 Validity of approach

Information from reconstructions of climate parameters and from simulation output together increase our understanding of past climates. The PAGES Hydro2k Consortium (2017) made recommendations for valid and appropriate comparisons of hydroclimate data from both sources of information. Here, we consider approximately the last 350 years by comparing both estimates to long instrumental data. We have to consider whether our analyes are appropriate in the sense of the recommendations concerning uncertainties, the properties compared, and the expectations underlying the comparison (PAGES Hydro2k Consortium, 2017).

The observational England-Wales precipitation data is a weighted composite of regional series which again are based on instrumental information. The input changed over time. Similarly, the reconstructions combine spatially distributed proxy, e.g., tree-ring width series into regional scale composite series (Cooper et al., 2013; Wilson et al., 2013) to maximise the common signal between different locations. On the other hand, the simulations are aggregations of various grid-point time series from the simulation output. We assume that the compositing and the aggregation have similar effects in removing local variability. In this respect, records from different sources are similar to each other and thus our comparison appears valid.

Explicit uncertainty estimates are only available for the reconstruction for East Anglia and only for a low-pass filtered version of the data (Cooper et al., 2013). Our results as well as the discussions of Cooper et al. (2013), Wilson et al. (2013), Rinne et al. (2013), and Young et al. (2015) emphasize that uncertainties for the reconstructions are potentially large and that even the relation to precipitation is not necessarily valid for parts of them. Similarly, uncertainties affect the simulations not only with respect to our domain choice but also with respect to the algorithms and parametrisations implemented for simulating precipitation in the regional climate model.

Considering the limitations of the simulation and the a priori known shortcomings of the reconstructions, questions may arise on the validity and robustness of our analyses. We argue that the transformation to standardized indices provides a sound basis for equivalence between the different precipitation estimates for subsequent comparisons of the distributional properties.

Then, we assume that the comparison becomes informative for changes over time between these distributions. While we cannot expect accurate or even approximate temporal agreement between time series from simulation output and observation based data on either interannual or multi-decadal time-scales because of internal variability, the transformation makes our comparison one of climatologies. Furthermore, one may assume that the evolution of percentiles and variability may be more consistent between the different data sets than the average conditions.

#### 5.3.2 Additional analyses

We find that the considered observations, reconstructions, and a regional simulation only show limited agreement in their representation of precipitation for a small regional domain covering southern Great Britain for approximately the last 350 years. Striking are the differences between the tree-ring width based reconstructions (Cooper et al., 2013; Wilson et al., 2013) and the observations, which again highlight the shortcomings of the two considered reconstructions (compare Young et al., 2015). It is

noteworthy that there are multiple periods where simulation output and reconstructions evolve oppositely. Possibly surprising is occasional temporal consistency in some of the measures between regional simulation and England-Wales precipitation data.

We performed similar analyses on a selection of the PMIP3-ensemble of global simulations (Schmidt et al., 2011). There, we see no commonalities between the different simulations or the simulations and the other sources of information in the analyses of precipitation distribution properties (not shown, see supplementary manuscript asset).

If we use different reconstructions, agreement between simulation and reconstructed precipitation does not necessarily increase, but differences between reconstructions and observations may be reduced (not shown, see supplementary manuscript asset). We use two different reconstructions based on $\delta^{18}O$. For one, we obtain the precipitation reconstruction by Rinne et al. (2013) for Southern England for the May to August extended summer season. Secondly, we use the isotope records for England and Wales by Young et al. (2015) and scale the composite against the observational England-Wales precipitation data. We follow the procedure described by Young et al. (2015) but for two seasonal estimates, the extended spring from March to July used in our main analyses and, following Young et al. for the summer season from June to August. See the supplementary manuscript asset for some details of the comparison to the summer scaling.

For the scaled data by Young et al. (2015), the most striking feature is again a notable difference in the percentiles prior to time windows approximately centered in the year 1850 compared to the later period. This feature resembles the behavior of the tree-ring width based reconstructions. While this may be due to the chosen calibration method and period, it appears more likely that there is a problem in the relation between isotopes and precipitation for this early period. Comparing the data to the extended spring observations, there is limited agreement for the dry percentile after this early period (not shown). For other periods, the moving window distributions show prominent inconsistencies compared to their observational counterparts. Comparing the data by Young et al. to the regional simulation also does not show any agreement.

The period covered by the data of Rinne et al. (2013) only shortly overlaps with the period of the observational data. For this overlap dry percentiles tend to agree with the observations but wet percentiles evolve oppositely (compare supplementary manuscript asset). The change in average precipitation for a reference year also agrees between both data sets for the time of overlap (not shown). Compared to the regional simulation output, evolutions tend to be opposite.

### 5.3.3 Implications of main results

Our analyses highlight the shortcomings of different reconstructions relative to observations. We also see that differences to observations may be comparable for reconstructions and simulations. Our methods also show that apparently the reconstructions and the simulations frequently evolve differently. This may signal that we indeed do not perform a valid comparison, that simulations may misrepresent forced responses, or, considering the reconstructions' relation to temperature, that the reconstructions do not fully relate to precipitation.

We expect disagreement between simulations and observations on some levels, not least because of differing influences of internal variability (see discussions below). More critical is the lack of consistency between reconstructions and observations. Most notably the reconstructions show unrealistically large changes in the cumulative probabilities represented by certain precipitation amounts (compare Figures 7 to 9). The reconstructions do not reliably represent the distributions in specific

periods. Plotting the anomalies for the observations and reconstructions (not shown) displays much stronger variability over the common period in the reconstructions compared to the observations and at times opposite trends.

One result is the inconsistency of the relations between temperature and precipitation in the data sets for the considered domains. Tout (1987) and Crhová and Holtanová (2018) both note the negative relation between temperature and precipitation observations for Britain. We find this only consistently in the simulation, and over the more recent period in the observations. The tree-ring width based reconstructions do not show any clear relation for the extended spring season. If we consider other seasons, the disagreement between data sets changes (not shown). The observations show consistently negative correlations for the summer season, and the scaled isotope data by Young et al. (2015) agrees quite well with the summer observations except for a large part of the 20th century when it shows a markedly weaker negative correlation (not shown). The simulation shows again generally stronger correlations compared to the other data in summer and shows some agreement with the observations in the industrial period since approximately the year 1850 (not shown). If we correlate the scaled isotope data to the temperature for an extended spring season from March to July, the correlations are quite similar to those for the larger domain simulation output but differ notably from the observations (not shown). The extended summer (MJJA) reconstruction by Rinne et al. (2013) agrees well with the respective observations in a consistently negative correlation (not shown). The relation is weaker for the reconstruction prior to the period of the Oxford precipitation observations (not shown).

Explanations for the different temperature-precipitation relations might be either physical inconsistencies within the simulations or a lack of physical relation between the temperature and precipitation records. A third possibility is that internal large-scale climate factors influencing the relation between both parameters evolve differently in simulation and reality. This implies a dominant influence of internal variability on the considered regional and temporal scales which we discuss in the next sub-section. Even though reconstructions and observations represent different regional domains, we tend to the inference that the disagreement between the observations and reconstructions suggests major shortcomings in the reconstructions, if we assume the observations to be the more reliable data set.

### 5.3.4 Internal vs. forced variability

If we look for temporal consistency among different source of informations, we assume that all sources of information reflect similar impact of the external climate forcing. Moreover, we then also believe the regional simulation to be skillful in representing the climate response to these conditions. We also have to be aware that internal climate variability may dominate even for large exogenous forcing (compare, e.g. Deser et al., 2012a). We can frame this as the question to what extend one can expect simulations and observation based data sets to reflect consistently these exogenous influences.

We assume that the transformation to distributional properties smooths out some of the temporal and structural differences from the different evolutions of internal variability expected between simulations and observational data. However, influences from low frequent climate modes may still have different phases in different data sets. In this sense, it is encouraging that the regional simulation shows some commonalities with the observed statistics.

The instrumental period overlaps with the industrial period of large anthropogenic climate forcing. Earlier in our period of interest exogenous forcing is potentially weak. However the period includes periods of relatively large variations in solar

activity like the late Maunder Minimum (~1645 to ~1715 CE), the Dalton Minimum in the early 19th century, and a period with relatively strong solar activity inbetween as indicated by sunspot numbers (Clette et al., 2014). Furthermore, a number of strong tropical volcanic eruptions occurred during this period, i.e. in ~1809 CE (unknown location), 1815 CE (Tambora), and 1835 CE (Cosigüina) (e.g., Schmidt et al., 2011).

Fischer et al. (2014) show that forced precipitation signals can agree in the CMIP5 21st century global projections. The lack of consistent relations between different data sets under purely externally naturally forced and internal variability on multi-decadal time-scales questions our ability to make dynamical inferences about hydroclimate variability of small regions.

A lack of an identifiable relation to the forcing does not necessarily imply that the underlying climate data are wrong but may simply suggest that internal natural variability dominates, e.g., oceanic, atmospheric, or coupled climate variability mask,
modulate, or counteract an external forcing influence. That is, the lack of consistent evolutions points to shortcomings of the data sources or an overwhelming influence of internal variability. The sporadic opposite behavior make the first more likely without negating the latter. That is, we interpret the opposite behavior as reactions to the forcing but different reactions of simulated and observed climate.

In this context, we have to emphasize that the regional simulation and its driving MPI-ESM-COSMOS simulation both use
variations of the total solar irradiance forcing that could be unrealistically wide. Furthermore, neither simulation includes a resolved stratosphere to account for potential UV-related top-down mechanisms (Anet et al., 2013, 2014).

Furthermore, since our regional focus is close to the western boundary of the domain of the regional simulation, we expect a rather strong influence of the dynamical evolution of the driving coarse-resolution simulation with MPI-ESM-COSMOS. Indeed, Blenkinsop and Fowler (2007) report a strong influence of the driving general circulation model on the representation
of drought in regional climate simulations in southern Great Britain.

Thus, while the regional simulation appears to present similar variations compared to the observations during some periods, it is unclear whether it does so for the right reasons.

### 5.3.5   Relation to dynamics

Our regional focus is a small domain. Thus, we should not expect simulations to agree with observations on the evolution of
regional climate parameters and even an ensemble may show diverse behavior since the influence of natural internal variability is large, e.g. in the case of the British Isles, the North Atlantic Oscillation (Gómez-Navarro et al., 2012; Gómez-Navarro and Zorita, 2013). Bengtsson et al. (2006) note the general importance of the storm track over the North Atlantic as a control on precipitation variability, and Dong et al. (2013) show this for the England-Wales summer season precipitation. Hall and Hanna (2018) study to what extent North Atlantic circulation indices explain precipitation in the United Kingdom. They note
a negative correlation of summer precipitation with indices representing jet-latitudes, which include the NAO. Blackburn et al. (2008) detail the large-scale influences, e.g., the wave-train pattern on the jet stream, on the flooding events in the UK in 2007. Earlier, Kington (1990) noted the strong relation between the England-Wales precipitation and Lamb's cyclonic British Isles weather type (compare, e.g., Lamb, 1950) in spring, while recently Matthews et al. (2016) emphasize the importance of the

high-frequent weather variability, i.e. cyclones, for seasonal precipitation amounts over the British Isles and particularly the summer season.

We note that Cooper et al. (2013) do not find any significant influence of the North Atlantic Oscillation (NAO) on precipitation or tree growth in East Anglia over the 20th century. They use the NAO as a measure of large scale influences on western European precipitation coherence.

Thus, internal climate variability is an integral expression of the circulation over the North Atlantic region. Differences in internal variability between models, observations, and paleo-observations may also include instabilities of dominant large scale patterns. That is, we cannot reject the idea that the relationship between regional climate and the large-scale circulation changed in the past. Lehner et al. (2012) describe the importance of such changes for inferring past states of the North Atlantic Oscillation from sparse proxy data. The importance of changes in the large-scale circulation becomes even more clear when considering the stability in centers of action in the North Atlantic sector over longer time-scales (Pinto and Raible, 2012; Raible et al., 2014).

That is, while the forcing history suggests notable variations and large-scale temperature records indicate an imprint of the forcing history on hemispheric and global temperatures, internal variability may dominate on smaller regional scales (e.g., Deser et al., 2012b). This is despite the fact that, e.g., the large scale storm track is indeed sensitive to solar (e.g., Ineson et al., 2015) and volcanic forcing (e.g., Fischer et al., 2007; Trouet et al., 2018). Considering the possibly large role of internal variability on regional scales and the limitations of simulations in representing regional scale precipitation, the occasionally consistent variations in precipitation distribution properties increase our confidence in forced changes.

### 5.3.6 Concluding remarks

In summarising, for the chosen regional domains, we do not find consistency among the various data sets. However, each of these data sets is associated with its own uncertainties, which put various caveats on the interpretation of the lacking consistency and its sources. Encouragingly simulations and observations appear to agree on certain features occasionally but maybe for different reasons.

## 6 Conclusions

This study pursued two goals. For one, we wanted to show that comparing precipitation in reconstructions, climate model simulations, and observations based on the Standardized Precipitation Index (SPI) over moving windows allows for the rigorous comparison of these data sets and extends the common set of tools for such analyses. Second, by using this approach, we studied the consistency of the various sources of information for precipitation variations in a small regional domain. We chose a domain in southern Great Britain and compared long-term trends, decadal variability, and the probability distributions for the period since approximately 1650 CE.

Fitting distributions over moving windows provides the opportunity to compare how the different sources of information represent various percentiles and moments of the distributions over time in the presence of varying external forcings. It further

allows to compare which percentile a reference precipitation amount represents over time; more loosely spoken, one compares how the probability of a reference precipitation amount changes over time.

For our specific study domain, we did not find any clear common consistency for precipitation signals among a regional climate model simulation, an observational data set, and two local domain reconstructions. The regional simulation shows only limited agreement with its observational target but less so with the reconstructions. The considered reconstructions indeed appear to be unreliable representations of the observational series. Relations between temperature and precipitation share some common co-variance on interannual time scales between the sources of information.

A further interesting result is the at times opposite evolution of the reconstructions and the regional simulations considering regional dryness and wetness. However, we cannot attribute it to the external forcing or to errors in either data source. The partial agreement between variability and dryness of the regional simulation and observations is encouraging but may be due to different processes in the respective data sources.

Generally, a dominant role of internal variability could explain the lack of consistency in standardised precipitation measures in the different data sets on the temporal and spatial scales we consider here; the relative role of the external climate forcing generally becomes weaker at smaller spatial and shorter temporal scales (Deser et al., 2012b). The lack of general consistency and slightly differing interannual relations between temperature and precipitation still require a closer look at the uncertainties of observations, the methods and input data of reconstructions, and dynamical and thermodynamical representations of regional climate in regional and global simulations.

**A supplementary asset for this manuscript will be deposited at https://osf.io/duyqe/.**

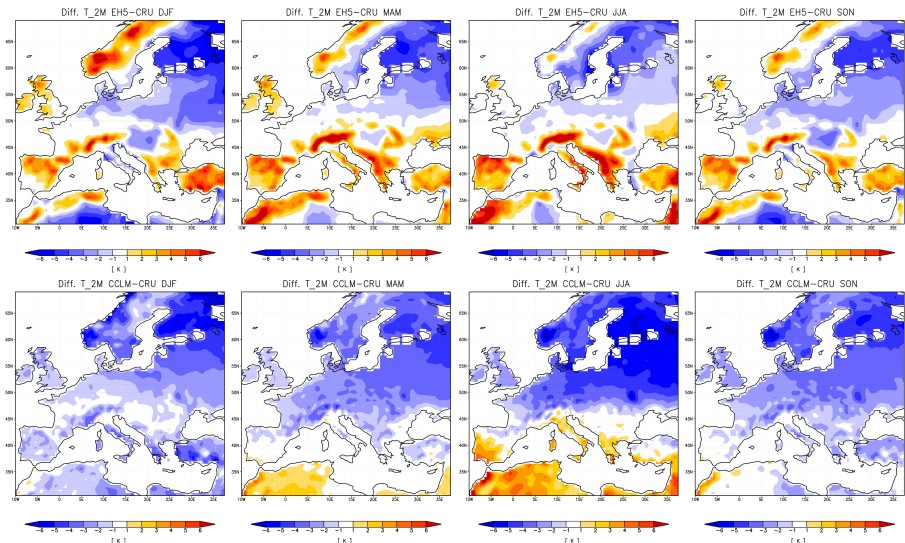

**Figure A1.** Top: Difference between the driving MPI-ESM simulation and the CRU data for seasonal near surface air temperature. Bottom: Difference for CCLM

*Data availability.* The Central England Temperature data is available from the Met Office, https://www.metoffice.gov.uk/hadobs/hadcet/.

The England-Wales Precipitation data is available from the Met Office, https://www.metoffice.gov.uk/hadobs/hadukp/ as are the subdivisions for South-East, South-West, and Central England.

Station data for Oxford, Kew Gardens and Pode Hole is available at, e.g., the Climate Explorer (http://climexp.knmi.nl/) of the Koninklijk
Nederlands Meteorologisch Instituut (KNMI).

The reconstruction data for Southern-Central England and East Anglia are available from the NOAA National Centers for Environmental Information at, respectively, https://www.ncdc.noaa.gov/paleo-search/study/12907 and https://www.ncdc.noaa.gov/paleo-search/study/12896.

Temperature and precipitation fields from the regional simulation with CCLM are available at http://doi.org/10.6084/m9.figshare.5952025 (PRIME2, 2018).
If deemed relevant for future work, we are going to provide the standardised data as well via a public repository.

## Appendix A: Evaluation of the simulation setup against the CRU-data

We shortly describe the performance of the COSMOS-MPI-ESM-CCLM-setup compared to the observational CRU-data (Harris et al., 2014; University of East Anglia Climatic Research Unit et al., 2017). We used version CRU TS 3.10, which has subsequently been superseded. The current version CRU TS 4.01 is available at http://doi.org/10/gcmcz3 with further informa-
15 tion also given at https://crudata.uea.ac.uk/cru/data/hrg/ (last visited 20 September 2018).

The mean climate of the driving COSMOS-MPI-ESM simulation is too warm for much of the British Isles, the Scandinavian Alps, northern North Africa, Iberia, the Alps, southern France, Turkey, and Greece for all seasons over the period 1951-2000 (Figure A1, top). It is generally too cold over the Baltic region, the eastern part of the model domain, the southern border of

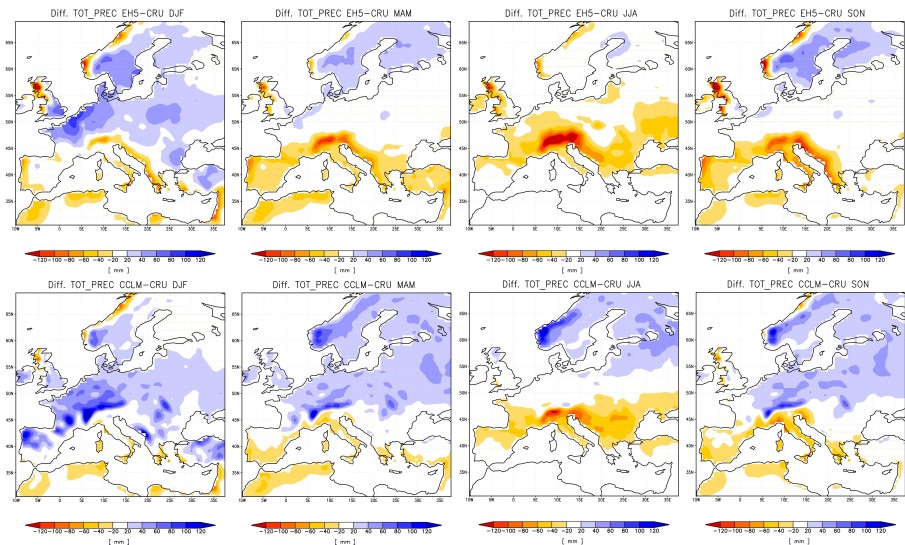

**Figure A2.** As Figure A1 but for the precipitation

the domain over Africa, and central Europe. High elevation and southern area warm biases frequently exceed 6K. Cold biases exceed 2 to 4K occasionally over northeastern Europe and at the southern border of the domain. We attribute these biases to some extent to the cruder representation of the European orography and, possibly related to that, to biases in the modelled atmospheric circulation. However, the specific choice of forcings may also influence the climatology.

In the regional CCLM simulation (Figure A1, bottom), warm biases for 1951-2000 are confined to the Atlas Mountains in all seasons and to the South of the domain in spring and summer. Cold biases are common otherwise and are largest over the Northeast frequently exceeding 3-4K.

For precipitation, summer is frequently too dry in central Europe in COSMOS-MPI-ESM and especially at the west coast of Scotland and in the Alps (Figure A2, top row). The southern domain is generally too dry in spring when Scandinavia is
slightly too wet. Coastal and mountainous regions as well as Iberia, Italy, and southern France are more likely to be too dry in autumn and winter. Scandinavia is also too wet in autumn. The COSMOS-MPI-ESM winter climatology is too wet over much of central, eastern, and northern Europe.

In CCLM, too dry conditions are generally confined to southern Europe and North Africa and areas affected by the storm track, i.e. the coasts of Scotland and Norway (Figure A2, bottom row). They extend to southern central Europe only in summer.
The climate is too wet in Scandinavia and northeastern Europe in most seasons. Large parts of Europe are too wet in all seasons except summer. Noteworthy is the excess precipitation at the northern flank of the Alps from autumn to spring. Part of these discrepancies are possibly attributable to a too zonal airflow outside the summer season.

In summarizing, the model presents a too strong latitudinal temperature gradient over the European domain. The annual cycle of temperature is apparently too strong in the South with warm biases in summer but cold biases in winter and it is slightly
too weak in the North with cold biases being stronger in summer than in winter. Similarly to temperature, the gradient in

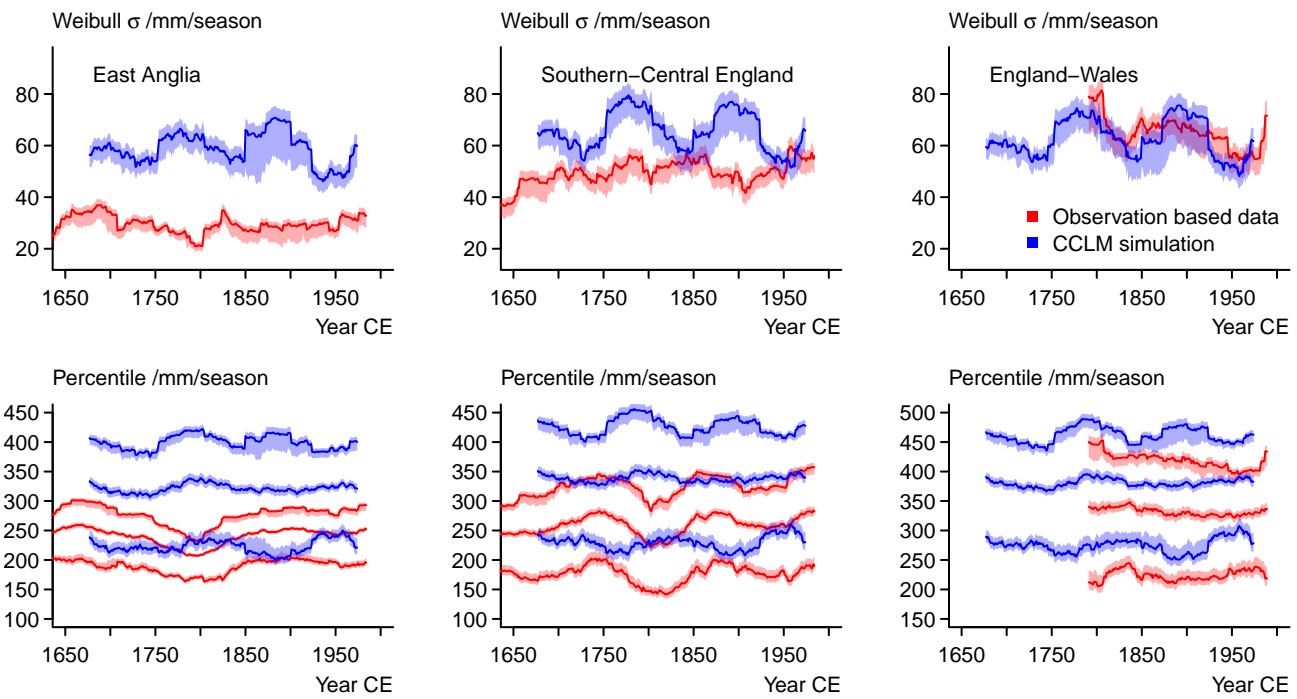

**Figure B1.** Visualisation of uncertainty of the distributional properties. We use a bootstrap procedure on running estimates. We resample 40 year samples a thousand times from moving 51-year windows. Units are precipitation amounts. Shading are 95% intervals, lines are medians. Top row: Weibull standard deviation. Bottom row: 93.3th, 50th, and 6.7th percentiles. Red: Reconstruction and observations. Blue: CCLM. The left column is for East Anglia, the middle column for Southern-Central England, and the right column for the England-Wales precipitation.

precipitation also appears to be too strong and the annual cycle amplitude differs between simulation and gridded observational estimates especially for Central Europe. Specifically, autumn to spring are wetter in the simulation while summer conditions differ only slightly or are too dry, which implies a weaker annual cycle compared to observations.

## Appendix B: Uncertainty of running measures

5   Figure B1 shows bootstrap estimates over thousand 40-year samples for each 51-year window. The estimates are for the running measures for reconstructions and observations for the three regions of interest (red) and the regional simulation (blue). The top row are Weibull standard deviations and the bottom row is for the percentiles.

    The Figure highlights that sampling variability is generally larger for the simulated data. Indeed sampling variability may render differences between periods non-significant. However, also the bootstrap distributions appear strongly skewed.

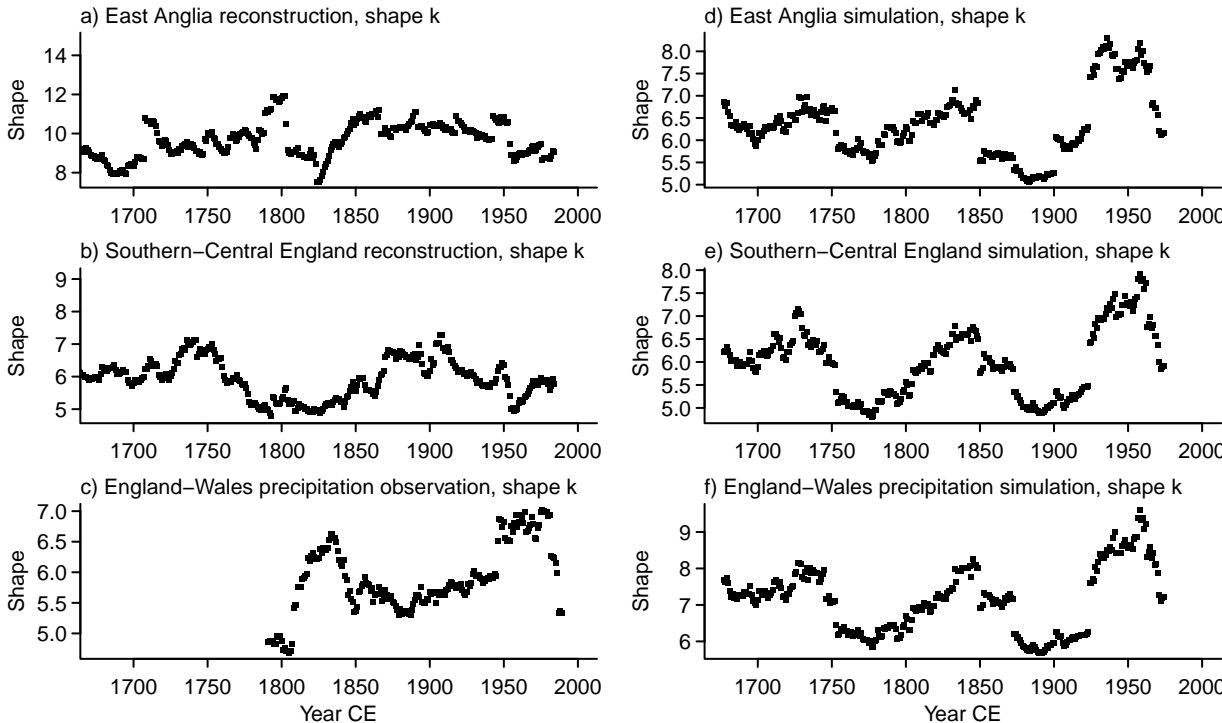

**Figure C1.** Evolution of the shape parameter $k$ for the Weibull distribution fits for the a) East Anglia reconstruction, b) Southern-Central England reconstruction, c) England-Wales precipitation observational data, d) East Anglia regional simulation, e) Southern-Central England regional simulation, f) England-Wales precipitation regional simulation.

## Appendix C: Distributional parameters

The Weibull distribution is a two parameter distribution with a scale and a shape parameter. See, e.g., Sienz et al. (2012), for more details and how the distribution compares to other distributions in computing the Standardised Precipitation Index.

Figures C1 and C2 present the shape, k, and scale, $\lambda$, parameters of our Weibull distribution fits for the reconstructions for

5   East Anglia and Southern-Central England, the observational England-Wales precipitation, and the respective time series in the simulation.

Results for the simulation show very similar evolutions among regions highlighting the homogeneity of the simulation data. There are also similarities between the two reconstructions. One could argue the shape parameters evolve similarly in observation and simulation.

10   The shape parameter determines the 'shape' of the distribution. In our cases, changes in this parameter are rather small (compare Figure C1). Nevertheless they can result in notably different widths of distributions for a specific data set over time. It is interesting that there is only small overlap between the range of shape parameters for the East Anglia reconstruction and all other series.

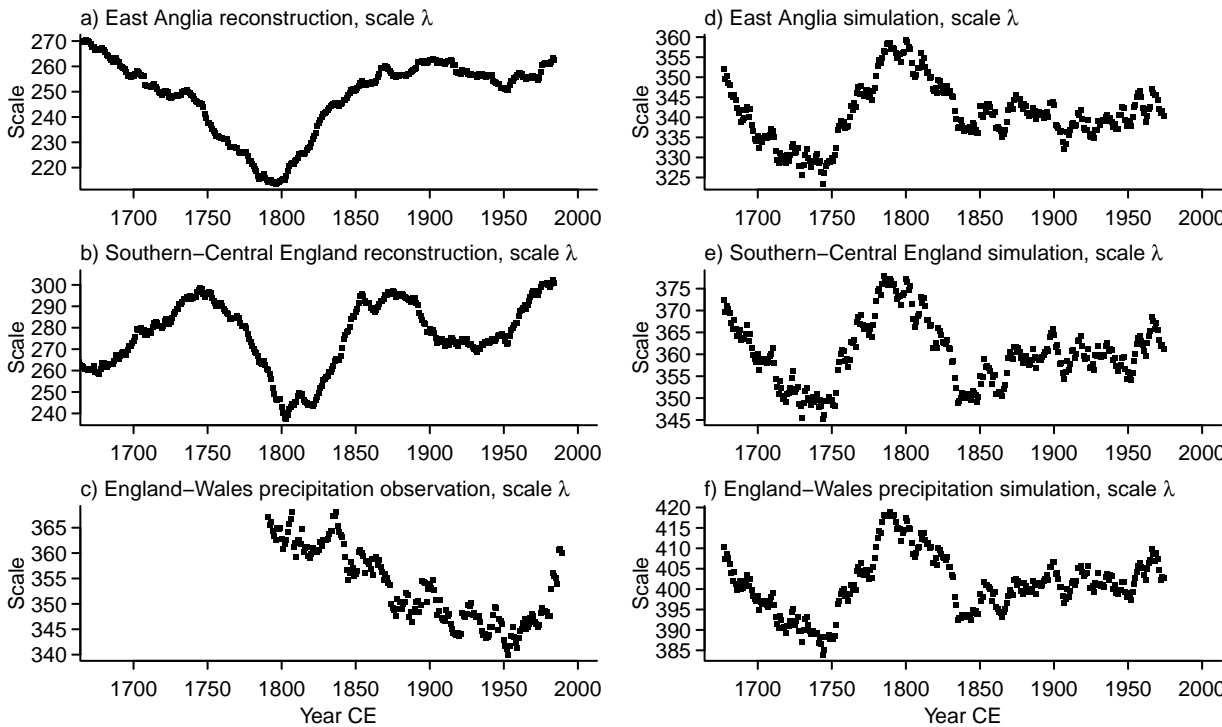

**Figure C2.** Evolution of the scale parameter $\lambda$ for the Weibull distribution fits for the a) East Anglia reconstruction, b) Southern-Central England reconstruction, c) England-Wales precipitation observational data, d) East Anglia regional simulation, e) Southern-Central England regional simulation, f) England-Wales precipitation regional simulation.

Larger scale parameters for a constant shape parameter result in a flatter distribution that extends further to larger values. Smaller values result in a narrower distribution with larger probability density at its peak.

The evolution of the shape parameter reflects, in our cases, the evolution of the skewness of the distributions (not shown). All distributions show negative skewness, and the amplitude increases with increases in the shape parameter.

5    Figure C3 shows the excess kurtosis over the period of interest. The most common feature for the different records is a negative excess kurtosis. Interestingly, the East Anglia reconstructions shows large positive values. The simulation data has a period with positive, or for the simulated England-Wales precipitation larger positive, values in the middle of the 20th century, and the observed England-Wales precipitation shows only negative excess kurtosis. The scaling of the kurtosis-axes for the reconstructions highlights that they show much larger values earlier in the last millennium (not shown, compare the

10    supplementary manuscript asset).

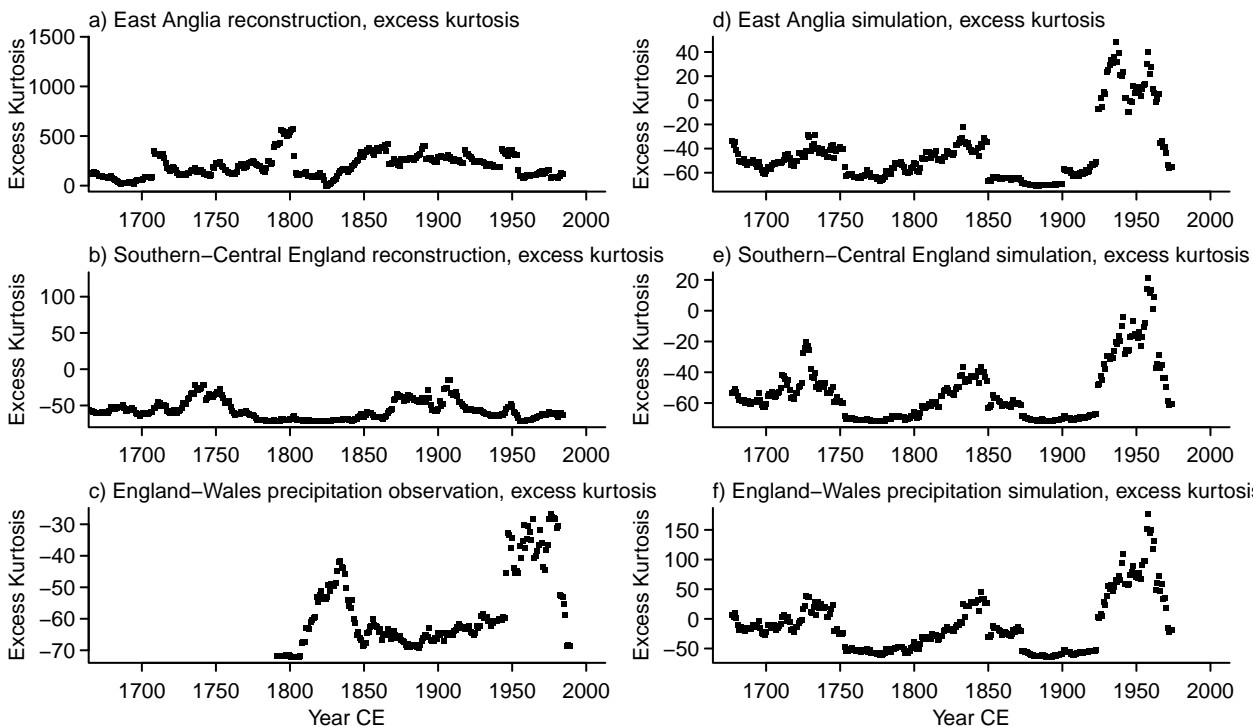

**Figure C3.** Evolution of the excess kurtosis of the fitted Weibull distributions for the a) East Anglia reconstruction, b) Southern-Central England reconstruction, c) England-Wales precipitation observational data, d) East Anglia regional simulation, e) Southern-Central England regional simulation, f) England-Wales precipitation regional simulation.

## Appendix D:  External code

This manuscript uses a number of external software-packages. File-manipulations used the Climate Data Operators (cdo, https://code.mpimet.mpg.de/projects/cdo/). Furthermore, the following R (R Core Team, 2018) packages helped in the work: gtools (Warnes et al., 2018), corrplot (Wei and Simko, 2017), ncdf (Pierce, 2015), VGAM (Yee, 2015), MASS (Venables and Ripley, 2002), nortest (Gross and Ligges, 2015), dplR (Bunn et al., 2018), zoo (Zeileis and Grothendieck, 2005), latex2exp (Meschiari, 2015), knitr (Xie, 2015), and rmarkdown (Allaire et al., 2018). Furthermore, RStudio (RStudio Team, 2016) was essential. The manuscript was prepared using the rticles-package (no reference available).

The SPI-code bases on work by Frank Sienz (e.g., Sienz et al., 2012). Christian Zang provided a Gershunov-bootstrap procedure (compare, e.g., Gershunov et al., 2001; Zang and Biondi, 2015) that we modified.

*Competing interests.*  The authors are not aware of any circumstances that might be seen as competing interests.

*Acknowledgements.* Funding in the projects PRIME2 and PALMOD (www.palmod.de) made the completion of this study possible. This study is a contribution to PALMOD, and to the PAGES 2k Network, especially its PALEOLINK project. We acknowledge the service of the Met Office Hadley Centre for Climate Change for providing the Central England Temperature and England-Wales Precipitation under the Open Government Licence (http://www.nationalarchives.gov.uk/doc/open-government-licence/version/2/), and of the NOAA Centers for

5    Environmental Information for providing the reconstruction data by Cooper et al. (2013) and Wilson et al. (2013). We acknowledge the SPI-code provided by Frank Sienz (e.g., Sienz et al., 2012). Christian Zang provided input for a computationally efficient Gershunov-test.

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
