# Peer review of "Inconsistencies between observed, reconstructed, and simulated precipitation indices for England since the year 1650 CE"

_Climate of the Past, 2018_

## Referee Comment (RC1) · Anonymous Referee #1 · 22 May 2018

Overall, I found this paper to be very confusing to read and also not surprisingly hard to follow. Also in this form I don't think that it is worth publishing. I think this is because it is attempting to do too much in a single paper. I would be more interested in one where the observations and reconstructions are compared, then a separate paper or a separate part of a single paper, where the climate simulations are compared. It seemed to me that every time there was something interesting, the discussion went on to a discussion of the models and initially a list of the models and all the necessary details about them in terms of refs/names/resolution/forcing etc. It would have been better if all this latter part was in a separate Appendix. The basic premise of the paper is that regional-scale precipitation (here for England and Wales) should show some impact of external

forcing, but it seems that not surprisingly that internal variability dominates. Maybe the authors should spend some more time looking at long observational series, and less time looking at climate model simulation output. There are long precipitation records for the England and Wales region (including also the Central England Temperature series) and they have been analysed for a long time (i.e. there is a vast literature on analyses of these series, that this paper doesn't consider at all). Figure 2 clearly realizes the seasonal nature of the reconstructions plotting seasons such as MAMJJ and JJA. Figure 1 though appears to look at annual averages for CET. So when in Figure 4 running correlations are shown for CET with precipitation observations, drought/precipitation reconstructions what season is being used. Is this CET for MAMJJ or JJA or is it annual CET? I couldn't decide what it is from the text or the captions. If it is CET annual then this is wrong. CET correlates with England and Wales rainfall in winter positively (warmer winters tend to be wetter) and inversely in summer (warmer summers tend to be drier). So relationships change with the season. Need to specify for every season what season is being used, otherwise people will assume annual like I did. These relationships ought to be captured by models, which is what I think you say, but this is buried in text somewhere else. This is another problem with the paper, that there appears little structure to it and the text doesn't flow in a logical order, and there is no summary at the end of the Introduction of what to expect in each of the subsequent sections. Some specific points 1. SPI. Using a distribution for this is discussed. Whatever is chosen, the parameters ought to be compared. Tree-ring based reconstructions generally explain only a portion of the variance, so these are likely to have a lot less variance than the observations. This issue needs to be discussed. Comparison of series at SPI doesn't let the reader see the effects of the differences in explained variance. 2. When you compare the reconstructions with England and Wales precipitation series in Figure 2, you seem to think that they will agree well. It is essential to look at how well SW England and also East Anglia compares with England and Wales. You can get the observed data here https://www.metoffice.gov.uk/climate/uk/summaries/datasets for periods since 1910. The correlations will not be as high as you imagine, partly because

East Anglia is dry and also how the England and Wales series is put together. See the brief discussion in Kendon and Hollis (2014). 3. There are odd bits of discussion almost on every page. On p7, why do you thing the late-18 the century dip in temperatures is due to the Laki Eruption? The references don't look at CET and the eruption did not put material in the stratosphere. I've assumed you're referring to CET as the paper is about this region, but the Laki eruption has been postulated as causing cooler weather in central Europe in 1784 and 1785, but as said this isn't very convincing at all (compared to say the eruption of Tambora in 1815). 4. You should state that all you expect with the models for this small a region is to get the precipitation amount right. You would need ensembles of runs to see if any of the low-frequency might agree. You seem to be expecting too much, or you need to explain why you're expecting as much as you are. 5. As stated the text is difficult to follow. Much of p10 comes into this category. The bottom line of Figure 2 shows Weibull standard deviations, but what does this mean? Surely this is showing what I was talking about in #2? The whole running numbers are confusing. It doesn't help putting too many coloured series on the already quite small plots. 6. The opposite evolution in East Anglia and SW England might be correct (p11)? You need to look at the observations to check this. There is an out-of-phase correlation between SE England and NW Scotland. 7. If series (p12) have the order of one degree of freedom, then what are you doing showing them. Parts of this page are very difficult to read and follow. 8. No seasons are given with Figures 4 and 5. 9. Trouet et al (2018) would have done better to have used the 300-year long instrumental records from the British Isles instead of going straight to tree-ring reconstructions. There are large variations across the British Isles with the size of the influence of the westerlies on precipitation amounts in the spring and summer. For example (p16) the NAO has no influence on East Anglian precipitation amounts in the winter half year. The NAO effect is much stronger on the western and northern areas of Britain, and it is mainly in the winter season. When you talk about spring/summer and the NAO are talking about the same NAO as in winter? It would be useful to discuss how the North Atlantic Jet that Trouet et al (2018) talks about relates to the NAO, if

it does? 10. P17 states that standardization of precipitation goes beyond comparing means and deviations. I'm not sure that you have shown anything other than just the means and SDs.

References Kendon M and D Hollis, 2014: How are UK rainfall-anomaly statistics calculated and does it matter? Weather 69, 37-39.

———————————————————

---

## Referee Comment (RC2) · Anonymous Referee #2 · 22 May 2018

While I think that this paper has merit and could provide interesting insight it is my view that it is not yet ready for publication. I encourage the authors to rethink the structure and layout of the paper and the key messages to be delivered. I think that such a paper would be welcomed by the field and of interest to the readers of the journal. But to reach a standard for publication significant work remains.

From the outset the specific aims of the paper are rather vague; the introduction section needs clearer structure. At the moment it jumps from one topic to the next without really unpacking where the state of knowledge it at in any aspect. The authors need to structure the introduction much more clearly, building the necessary context for the

reader to understand what the aims are and the summary of information necessary to move to the next stage.

If the focus is on the British Isles why just use the EWP series and not the Island of Ireland monthly series from 1711 or the Scottish regional series. I realise the latter is shorter, but to talk of the British Isles and not use the other available regional series is confusing. Murphy et al. (2018) cited in the introduction show that CET is also strongly correlated, at least at decadal scales with the Irish series.

Why did the authors choose these tree ring reconstructions? To the best of my knowledge these are based on ring width reconstructions which have been shown to be less reliable for precipitation. Why not incorporate the oxygen isotope reconstructions done by Rinne et al. (2013) for southern England. Indeed in their discussion, if i recall correctly, they identify interesting points of departure from both EWP and Kew precipitation series for the summer months. Again in providing this suggestion as I am reading it is not clear what the time focus is of the paper – spring/early summer, spring?? The study design needs clearer thought, signposting and explanation.

Regarding the selection of ensemble members from model reconstructions, why not use the entire ensemble? In the next paragraph it is noted that the selection is rather arbitrary and it is assumed that the domain sufficiently represents EWP domain. Some kind of table to help the reader interpret the different forcings used would be helpful.

The use of the SPI to investigate the 6.7 and 93.3 percentiles is a very stringent test of models and reconstructions is it not? The EWP is essentially a composite series and extremes are likely smoothed out. Also, is it a fair ask to expect climate model reconstructions to be able to represent these, especially if not employing a large ensemble? I am only asking out of curiosity here and would like to be informed of how stringent the comparison you are setting up is.

Any bias correction applied to the models? Does SPI negate this?
Results presented in the methods section need to be moved.

The paper is badly let down by plots that are very hard to decipher and methods applied that are not appropriately, or sometimes not at all, explained in the methods section.

Fig 1 – no detail of the types of smoothing applied covered in the methods. What is a 'first impression', not a scientific term. What CET time step is the smoothing applied to? Monthly or annual series. Why not plot as an ensemble rather than 11 sub plots? Line types in legend do not match the plots. Use of sunspot data is not covered in the data section so far as I recall.

A table detailing the various data sources compiled is badly needed.

The use of differing periods is confusing, how can this be comparative – which is the primary aim of the paper.

Please think about presenting results in a clearer way. I literally spent hours trying to figure out what the figures were showing and in many aspects am no clearer.

There needs to be a more systematic approach to this work in terms of presentation and some sub sectioning in the results and discussion to help the reader.

The title of the paper concerns precipitation. It is confusing to start the results off with temperature.

I find it next to impossible to interpret the caption of Figure 2.

It is difficult to comment in much depth on the nature of the results and the points made in discussion and conclusion given how difficult it is to decipher what was done.

Authors need to revise the structure of the paper to systematically consider the inconsistencies of interest.

---

## Referee Comment (RC3) · Anonymous Referee #3 · 12 Jun 2018

Summary: The manuscript involves a comparison of climate model simulations with an ensemble of global and one regional model to long observationally-based records and two paleoclimate reconstructions. Little consistency is found between time histories of these records, suggestive of a large role for internal atmosphere-ocean variability. Importantly, while there is little agreement between the characteristics of the model simulations and the observationally-based records, these differences do not appear to be systematic across models and cannot be explicitly linked to model bias. Likewise, there appears to be even less agreement between the characteristics of the observationally-based records and the reconstructions. Together this work is consistent with mounting evidence that regional hydroclimate is largely "unforced".

[Figure]

General Remarks: While the manuscript is interesting and highlights some important results, it is at times unclear what should be taken away from the results. This is, in part, an issue with the introduction and a refocused introduction that clearly describes the motivations and goals of the study would greatly improve the manuscript. Below are a number of specific and more general comments.

Page 1, Line 8: and in the standard deviations seems a weird statement.

Page 1, Line 18: add "of" before "whether".

Page 1, Line 19: what is meant by requires consistency?

Page 1, Line 21: suggest changing to "over approximately the last 350 years".

Page 2, Line 2: suggest removing "in particular".

Page 2, Line 6: change "base" to "basis".

Page 2, Line 10: change "compare directly" to "directly compare".

Page 2, Line 12: Cooper and Wilson et al. are the reconstructions. I would be careful here and throughout with the semantics of "data".

Page 2, Line 16: You argued in the paragraph above that you do not want to use gridded reconstructions. I understand that this paragraph is addressing a new issue but the reference to the OWDA thus seems unusual here. In general, this paragraph does not seem necessary. I might instead start at the beginning of the next paragraph and add a statement at the end of that first sentence saying that you are doing the standardization to make the reconstructions directly comparable to SPI.

Page 2, Line 24: Suggest changing "their data" to "the utilized archives".

General comment: A lot of the above reads much more like a methods section than an introduction. I suppose this is more of a personal preference but the paper might be more impactful with a standalone introduction that does not include this methodological

information.

Page 2, Line 24: The sentence about Murphy et al. (2018) feels out of place. I would try to tie this into the paragraph above or remove it.

Page 2, Line 28: Suggest removing "than in periods that are more recent".

Page 2, Line 29: Suggest splitting the sentence after the Maunder Minimum dates. I would then reword as: "Instead, they generally start around the late 18th century, when sunspot numbers indicate a period of relatively strong solar activity (Clette et al., 2014), and thus also include the transition. . ."

Page 2, Line 35: Suggest changing "in European subdomains" to "across Europe".
Page 3, Line 1: Change "extend" to "extent".

Page 3, Line 10: This sentence is long and the second half I had trouble understanding. Perhaps you could split this up into two sentences and expand on the point that you are trying to make in the second half of the sentence.

Page 3, Line 20: Suggest "using the global model ECHO-G for boundary conditions" instead of "externally forced". I am also not sure what this part of the sentence means: "and reconstructions over larger regional domains."

Page 4, Line 1: Suggest changing "and the simulation data representing" to "and simulations that often represent".

Page 4, Line 2: Suggest changing "evaluation" to "comparison".

Page 4, Line 17: Change "allows comparing" to "allows for the comparison of".

Page 4, Line 19: Change "allows evaluating and comparing" to "allows for the evaluation and comparison of".

Page 4, Line 22: Suggest changing "extends the available metric for assessing the agreement in" to "allows for the rigorous comparison of".

Page 4, Line 23: Suggest changing "not only for periods without but also with" to "for periods both with and without".

General comments on introduction:

I am unsure about the relevance of short-term (decadal) relationships between temperature and precipitation with those expected as a result of climate change (first two sentences of the introduction). The relationship between hydroclimate and temperature at the end of the 21st century in climate models is largely due to evaporative demand, which has a first order impact on water storage but not necessarily on precipitation. These changes are also very large in magnitude, and co-occurring with large magnitude changes in plant physiology, making deeper-time paleoclimate comparisons more appropriate for evaluating climate models (e.g., Scheff et al., J. Clim., 2016). I do not think this precludes such analyses being useful, I am just unsure of using the relationship between temperature and precipitation with an eye towards climate change as the motivation.

I would be careful with the semantics of the word data to make sure that things are as clear as possible. Likewise, I would refer to reconstructions, observations and simulations each with a single consistent term. This applies to the entire manuscript.

The introduction bounces around a lot, with quite a bit of methodology (see general comment above). I think that as cast it will leave the reader uncertain about the motivations and goals of the study. I suggest that the authors revisit the introduction with an eye towards clarity.

I made an effort to make grammatical edits in the introduction but likely missed some. I will not be able to make this effort in subsequent sections but suggest that the authors revisit the manuscript with an eye towards grammar and syntax. It might be worth explicitly outlining how what you are doing here is different from Gómez-Navarro et al. (2015). Along with what is described the methods there would appear to be quite a bit of overlap.

Page 4, Line 28: Change to "in the form".

Page 5, Line 15: Suggest adding "In particular," at the start of this sentence to link it to the previous sentence. Suggest also changing "different means" to "systematic differences in the values of".

Page 5, Line 16: Suggest "While model-biases may also contribute to these differenences,..." and change "bias" to "source of differences".

Page 5, Line 17: I doubt it matters but why the different domain here?

Page 5, Line 30: Change "to include" to "the inclusion of".

Page 6, Line 5: Change "allows to compare" to "allows for the comparison of".

Page 6, Line 19: Change "allows considering the changing amount of precipitation" to "allows for a robust quantification of changes in precipitation amounts between subsequent periods, for instance ".

Page 7, Line 2: Remove "just".

Page 7, Line 3: Add "the" before "time series".

General comments on methods:

The half-degree simulations are course resolution for a regional climate model. At least one of the last millennium simulations analyzed is one degree (CCSM4), how much added information do we expect from a regional simulation at this course resolution and what physical processes is it capturing to provide that information?

Page 7, Line 13: Change "tentative" to "qualitative".

Page 7, Line 20: Suggest change the last sentence to "This is likely to also impact our analyses of precipitation".

Page 7, Line 32: What is the European domain?

Page 8, Line 2: Suggest removing the first sentence.

Page 8, Line 5: Suggest changing "representations" to "time series".

Page 8, Line 7: Suggest removing "but the Southern-Central England data enters it later".

Figure 2, caption: Why call the Southern-Central England record SW England in the legend?

Page 10, Line 23: Change "allows evaluating" to "allows for the evaluation of".

Page 10, Line 24: Change "gliding" to "sliding".

Page 10, Line 25: Suggest removing "partially".

Page 10, Line 27: Change sentence to read "The moving window transformations show the percentiles represented by a given amount of precipitation over time (Figure 3)."

Page 12, Line 1: Suggest changing "We pointed above at" to "In the previous sections we described".

Page 12, Line 6: Suggest changing "gliding" to "sliding".

Page 12, Line 11: Suggest combining these two sentences.

Page 12, Line 12: Suggest changing "Considering" to "In".

Page 12, Line 15: Suggest removing "correlation".

Page 12, Line 20: Suggest changing "highly" to "strongly".

Page 12, Line 21: Why "CET" here and not elsewhere?

Page 12, Line 31: Change "very low frequent variability" to "low frequency variability".

Page 15, Line 8: Again why the use of "CET" here and not elsewhere?

Page 15, Line 15: Why just atmospheric circulation when coupled variability can also

10.5194/cp-2018-27
2018
Clim. Past Discuss.

do this?

Page 15, Line 24: I found this paragraph difficult to understand. The final sentence is seemingly important but I was unclear on what it means. Likewise, I would clarify what is meant by unfortunate earlier in the paragraph.

Page 16, Line 7: Suggest changing "appears" to "is".

Page 16, Line 23: While this is true, it is unclear how it relates to the other discussion.

Page 17, Line 19L Change "source" to "sources".

––––––––––––––––––––––––

---

## Author Comment (AC1) · 15 Jun 2018

Dear reviewers, dear editor,

Thank you for your candid and helpful judgment, comments, and suggestions.

Here, we provide an initial reply to your reviews. More detailed replies will follow. For the time being, we would like to outline our plans for a revised manuscript with this initial reply addressing the basic concerns raised in the reviews.

All of your comments made it obvious that we have to substantially increase the clarity and improve the structure of the manuscript. This relates to the format and content of

the introduction and the method sections as well as the aims, scope, expectations, and the conclusions of the manuscript as a whole. It also includes a clearer presentation of the results in terms of plots and structuring.

Regarding the aims of the study we would like to already clarify that our main points are a) motivating the advantages in using the SPI for comparing various sources of information in paleoclimate research and b) finding that the sources of information lack consistency for the case of a rather small domain on the British Isles.

Remediating the structural and other weaknesses of the manuscript likely requires more than just a simple restructuring, i.e. it needs a thorough rewriting of at least the introduction and the methods section. We also have to more clearly delineate our work from some previous papers and more clearly motivate the relevance of some of our work.

A number of comments relate to the arrangement, the amount, and the type of analyses presented and data included. For the moment, we plan to downsize the role of the temperature data, the analyses on the PMIP3-simulations, and the correlation analyses. Potentially, we are going to move the analyses on the PMIP3-simulations and the temperature data completely to an appendix or a supplementary document. At the moment, we plan to present the information about the parameter fits in some kind of auxiliary material as well.

Regarding analyses on additional data, we have to check the robustness of our results by also considering the Met Office's sub-divisional precipitation series included in the England and Wales data, and the long observational series for Kew Gardens and Pode Hole. It is open whether this will result in including the additional analyses in the manuscript.

You suggested to include more long observational series and additional reconstructions. On the other hand you noted that we possibly tried to present too much in a single manuscript. Our aim is not a comprehensive analysis of all available data for the

British Isles. Our intention is to show for a small region the (in)consistencies between the various information sources. For the moment, we do not plan to include additional reconstructions or regions on the British Isles but rather to optimise our presentation of our chosen focus.

We will provide separate responses to all your detailed comments later in the coming weeks.

We are convinced that these proposed modifications are going to strengthen the manuscript and hope they reflect your intentions and conform to your comments.

On behalf of all co-authors,

Yours sincerely,

Oliver Bothe

---

## Author Response (AR1)

Dear editor,

Thank you for your comments, suggestions, and recommendations.

In the following we address each of your major points separately. Your comments are in red font, our replies in normal font.

1. thanks very much for addressing the questions/critics/suggestions of the reviewers. All reviewers mention similar major points with regard to the clarity of the introduction, aim of the study, the presentation of the results, critical aspects related to the choice of instrumental data, reconstruction and models, their interpretation, visualisation of the results as well as the conclusions. Thus, it should be straight forward to revise the manuscript accordingly in agreement with the suggested changes of the authors.

We have rewritten and rearranged the manuscript. We changed the visualisation of results and added additional Figures. We hope this clarifies the presentation and addresses the points made by referees and editor.

2. The authors use SPI as the measure for their analysis. As it is only based on precipitation, I agree that analysis that use temperature do not need to be shown.
   a) The revised version thus should include more studies that deal with precipitation and as suggested by the authors in their last section of reviews, incorporate additional precipitation/drought sensitive information.
   b) This would mean that the revised version provides a comprehensive analysis of all available precipitation and drought information from parts of the British Isles.

We reduce the discussion of temperature to a minimum, but we would like to point out that the rationale for including temperature was not its influence on the drought index but rather the physical understanding of precipitation anomalies, for instance, whether there is a  link to cloudiness variations.

   a) The revised version discusses a number of additional studies on precipitation over the South of Great Britain. We use additional data in form of instrumental series, of observational indices, and discuss shortly additional reconstructions. We do not include the analyses of these series in the manuscript, as these are not all in the public domain and we are not willing to compel the original authors to publish their data because of our manuscript.
   b) Thus, we are quite comprehensive for the south of Great Britain. We still do not include field reconstructions and we do not include other parts of the British Isles.

3. The (in)consistencies between the various information sources need to be made more clearly as the reviewers suggested.

We hope that the rewriting of the manuscript and the additional discussions clarify what we mean by consistency, what we mean by inconsistency, and which consistencies and inconsistencies we find.

4. It would be important to explain also in more detail why SPI is used and not other drought related measures. SPI is a specific index for meteorological drought with strengths and limitations that need to be discussed (see information for instance in: http://climatedataguide.ucar.edu/climate-data/standardized-precipitation-index-spi About the choice of an appropriate drought index from a model point of view, the authors might refer to Raible et al. (2017) on "Drought indices revisited-improving and testing of drought indices in a simulation of the last two millennia for Europe", Tellus, 69, 1287492

Introduction, methods section, and discussions now include dedicated paragraphs justifying our choice of the SPI, describing the SPI's shortcomings and advantages, and describing its appropriateness for the region of interest.

5. For instrumental data/reconstruction data the authors might do a detailed screening on the literature for this area and then choose.

We describe now clearly our choice of instrumental data and add these to the initial analyses. We further discuss them.

6. Further, the use of RCM data for comparisons, interpretation, comments that have been risen all reviewers. This point needs to be more clearly addressed, what is the purpose of showing RCM analysis in comparison with instrumental/reconstruction data and how do the analysis go beyond the current state of the art. The authors mention that they will make this point more clear and also move parts in a SOM.

Introduction, methods section, and discussions now include dedicated information and comments on why and how we use the Regional Climate Simulation. The global simulations have been removed from the main manuscript and relocated into a supplementary document.

7. In this context one open issue refers to the suggestion of reviewer 1 to write two different papers rather than implementing everything in one paper, thus removing for instance the model/data comparison and leave it for a new publication. I leave the decision up to the authors, but any decision to include models or exclude them should be clearly explained.

We decide against the option to write two papers on the topic. We regard the comparison of all three sources of information as one of the major points of this manuscript. One point here is that the comparison between reconstructions and observational data is already done in the original publications, although not in terms of SPI. Secondly, as the added value of RCMs is according to your comments still up for discussion, we think including them here may help to show their value. The manuscript as a whole is certainly quite extensive, but we think it forms a unity that would be difficult to split in two parts. We are considering to extend on additional reconstruction data in a complementary manuscript.

8. I agree with Reviewer 1 and 2

a) that have concerns using the tree ring reconstructions, either as they may not reflect precipitation or the season under consideration

b) or that the Trouet et al. jet reconstruction is not an appropriate measure for circulation purposes. With respect to circulation analysis, for the past centuries there are monthly to seasonal large scale SLP and Z500 reconstructions that are based on instrumental pressure and ship log books information. They are more suitable and trustworthy than natural proxy reconstructions and should be used instead.

a) We shortly discuss two isotope based reconstructions.

b) Given the overall length of the manuscript, we decided against including reconstructions of atmospheric dynamics and their comprehensive discussion except for the part where we discuss the literature more extensively.

9. I think it is a valuable comment by reviewer 2 concerning the Rinne et al paper. I am sure that the authors would provide the data for analysis.

We obtained the data and shortly discussed it.
* * *
We upload the Supplement now with the manuscript. It will be ultimately deposited at https://osf.io/duyqe/.

Below you may find our final point-by-point reply to the reviews as already posted in the discussion forum on 16 July 2018, a list of relevant changes to the manuscript, and a marked-up version of the manuscript.

Thank you for your helpful comments again.

On behalf of the authors,

Yours sincerely

Oliver Bothe

Clim. Past Discuss.,
https://doi.org/10.5194/cp-2018-27-AC2, 2018

[Figure]

Once more, thank you for your candid and helpful judgment, comments, and suggestions.

Let us start with a preliminary note. We will follow the advice by the editorial office and are not going to prepare a revised version right now but rather wait for the editor's recommendations. While a structure of a potential new version is clear and presented below, our replies to the referees' comments depend on whether the point in question

will be included in a new manuscript.

Next we shortly repeat and slightly extend on our initial reply. Detailed responses to all comments follow below.

Intended changes in document:

Regarding the aims of the study we would like to point out once more that we a) aim to motivate the advantages in using the SPI for comparing various sources of information in paleoclimate research on precipitation and b) find for the case of a rather small domain on the British Isles that the sources of information lack consistency.

Our aim is not a comprehensive analysis of all available data for the British Isles. Our intention is to show for a small region the (in)consistencies between the various information sources. For the moment, we do not plan to include additional reconstructions or regions on the British Isles but rather to optimise our presentation of our chosen focus.

Your comments made it clear that we have to increase the clarity and improve the structure of the manuscript. Therefore we will rewrite abstract, introduction, methods section, and conclusions. This hopefully clarifies the motivation, the expectations, the aims, the methods, and the implications of our results. We will also improve the visual presentation. The results-section will require a profound revision, too, considering that we will likely add new analyses and remove some of the original analyses.

Regarding the results, the new version will concentrate on the analyses of the distributional precipitation properties, i.e., the SPI. We will add a comparison to further regional observational information sources. A manuscript asset, e.g., an appendix, is going to present shortly the Weibull distribution fits.

In turn, we will minimise the comparison between precipitation and temperature data and overall the analyses of temperature data. We aim also to regard the global simulations only in passing. Both parts will be moved to manuscript assets but these will be

truly supplementary to the scope of our manuscript.

Detailed responses to all your comments follow below. Referee comments are put in red font and our replies in blue font. Intended changes in the document follow in default font.

**0.1 Referee 1:**

Overall, I found this paper to be very confusing to read and also not surprisingly hard to follow. Also in this form I don't think that it is worth publishing.

Author reply: Thank you for your honest evaluation. We aim to remedy your concerns in a new version.

Intended changes in document: We intend to rewrite abstract, introduction, methods section, and conclusions. This hopefully clarifies the motivation, the expectations, the aims, the methods, and the implications of our results. We will also improve the visual presentation.

I think this is because it is attempting to do too much in a single paper. I would be more interested in one where the observations and reconstructions are compared, then a separate paper or a separate part of a single paper, where the climate simulations are compared.

Author reply: We aim at this point to mostly remove the global models from the manuscript. The discussion of the regional simulation is going to be included in a new version of the manuscript.

Intended changes in document: The discussions and analyses of the PMIP3-ensemble is moved to a purely supplemental manuscript asset.

It seemed to me that every time there was something interesting, the discussion went on to a discussion of the models and initially a list of the models and all the necessary details about them in terms of refs/names/resolution/forcing etc. It would have been better if all this latter part was in a separate Appendix.

Author reply: We are not quite sure what you are referring to. However, descriptions of the PMIP3-ensemble will be removed from the main manuscript.

Intended changes in document: Remove descriptions of the PMIP3-ensemble.

The basic premise of the paper is that regional scale precipitation (here for England and Wales) should show some impact of external forcing, but it seems that not surprisingly that internal variability dominates. Maybe the authors should spend some more time looking at long observational series, and less time looking at climate model simulation output.

Author reply: Most of the analyses are on reconstructions and observations. We will add comparisons on additional Met Office data. We will discuss our expectations more. Indeed our basic premise is shortly described as: we need consilience of evidence from all sources of information to reach a robust understanding of past and future climate variability and climate changes. This also involves external forcings. Agreement about forced and unforced signals may signal consilience. Internal variability is likely to dominate the mean signal on regional scales, the SPI-transformation allows to compare quantiles of precipitation data more easily as well as other precipitation distribution properties.

Intended changes in document: We will add discussions on the data we use by comparing to additional data. These are the subdivisions of the Met Office Hadley Centre England-Wales precipitation data, i.e. the data for South West, South East, and Central England. Additionally, we use the instrumental precipitation data from Kew Gardens and Pode Hole. We will also specify our basic premises more clearly.

There are long precipitation records for the England and Wales region (including also the Central England Temperature series) and they have been analysed for a long time (i.e. there is a vast literature on analyses of these series, that this paper doesn't consider at all).

Author reply: We are not sure to which papers you refer specifically - or how they refer to the current analyses. We will however screen again respective databases in case we have overseen high quality and long data series over our region of interest and papers

discussing them.

Intended changes in document: Some additional references may be added.

Figure 2 clearly realizes the seasonal nature of the reconstructions plotting seasons such as MAMJJ and JJA.

Figure 1 though appears to look at annual averages for CET. So when in Figure 4 running correlations are shown for CET with precipitation observations, drought/precipitation reconstructions what season is being used. Is this CET for MAMJJ or JJA or is it annual CET? I couldn't decide what it is from the text or the captions. If it is CET annual then this is wrong. CET correlates with England and Wales rainfall in winter positively (warmer winters tend to be wetter) and inversely in summer (warmer summers tend to be drier). So relationships change with the season. Need to specify for every season what season is being used, otherwise people will assume annual like I did. These relationships ought to be captured by models, which is what I think you say, but this is buried in text somewhere else.

Author reply: All analyses use MAMJJ except when explicitly stated that it is JJA. We will clarify this point.

Intended changes in document: The results section will be more clearly formulated, Figure captions revised, and connections between text and Figures optimised. We aim to specify the season for every analyses described in the text.

This is another problem with the paper, that there appears little structure to it and the text doesn't flow in a logical order, and there is no summary at the end of the Introduction of what to expect in each of the subsequent sections.

Author reply: We will try to remedy this structural issues.

Intended changes in document: The structure will be revised. We do not plan to add a redundant summary of following contents at the end of the introduction.

Some specific points

1. SPI. Using a distribution for this is discussed. Whatever is chosen, the parameters ought to be compared.

Author reply: As clearly stated in the manuscript, we fit and use Weibull distributions. We will shortly present the parameters.

Intended changes in document: Manuscript assets are going to present the parameters of the Weibull distribution fits.

Tree-ring based reconstructions generally explain only a portion of the variance, so these are likely to have a lot less variance than the observations. This issue needs to be discussed.

Author reply: We will discuss this in more detail in the new version.

Intended changes in document: We are going to discuss the variance issue and how the original authors of the reconstructions rank this issue.

Comparison of series at SPI doesn't let the reader see the effects of the differences in explained variance.

Author reply: We are unsure on the point raised by the reviewer. The pure reconstruction series do not show this either in our original Figure 2.

Intended changes in document: As mentioned above, we will be more clear about the variance issue.

2. When you compare the reconstructions with England and Wales precipitation series in Figure 2, you seem to think that they will agree well. It is essential to look at how well SW England and also East Anglia compares with England and Wales. You can get the observed data here https://www.metoffice.gov.uk/climate/uk/summaries/datasets for periods since 1910. The correlations will not be as high as you imagine, partly because East Anglia is dry and also how the England and Wales series is put together.

See the brief discussion in Kendon and Hollis (2014).

Author reply: As you may have seen, we obtained data from the Met Office homepage. We will clarify these points. England-Wales does correlate very highly with South-West, South-East, and Central England on interannual time-scales and also for lower resolution data. The reconstructions are much less related to the observational sets on both scales. We do not see the relevance of Kendon and Hollis (2014) for this discussion.

Intended changes in document: We will include a more extensive comparison to the observational data series.

3. There are odd bits of discussion almost on every page. On p7, why do you thing the late-18 the century dip in temperatures is due to the Laki Eruption? The references don't look at CET and the eruption did not put material in the stratosphere. I've assumed you're referring to CET as the paper is about this region, but the Laki eruption has been postulated as causing cooler weather in central Europe in 1784 and 1785, but as said this isn't very convincing at all (compared to say the eruption of Tambora in 1815).

Author reply: We will clarify these points. Especially, we will discuss why we think the high latitude eruption of Laki could have an influence on European climate. You state yourself that the effect has been postulated, which by itself warrants inclusion of this date. However, we have to discuss in more depth how likely the eruption may have had an impact on European and British Isle temperatures in an extended spring season.

Intended changes in document: We will discuss the inclusion of this date if the new version includes this discussion.

4. You should state that all you expect with the models for this small a region is to get the precipitation amount right. You would need ensembles of runs to see if any of the low-frequency might agree. You seem to be expecting too much, or you need to explain

why you're expecting as much as you are.

Author reply: We will clarify our expectations. Indeed we absolutely do not expect to get the amount right. This is one reason why we think the SPI may help. If there is a forced response we may see it in the mean. Assuming the internal variability dominates the mean series, the SPI additionally allows to have a look at other properties of the precipitation distribution to see whether these may show a signal. Indeed, the suite of PMIP3-simulations and our regional simulation represent an ensemble.

Intended changes in document: We will more clearly discuss our expectations, why we think we do not need an ensemble of simulations, and why we in the end remove the global simulation ensemble.

5. As stated the text is difficult to follow. Much of p10 comes into this category. The bottom line of Figure 2 shows Weibull standard deviations, but what does this mean? Surely this is showing what I was talking about in 2? The whole running numbers are confusing. It doesn't help putting too many coloured series on the already quite small plots.

Author reply: We aim to provide a new version which is easier to follow. We will try to clarify all these points. The Weibull Standard deviation is the square root of the Weibull distribution variance as, e.g., presented at http://mathworld.wolfram.com/Variance.html (Weisstein, Eric W. "Variance." From MathWorld–A Wolfram Web Resource) or in a number of textbooks.

Running numbers allow displaying easily the changes in the distribution properties. We aim at improving the visualisations of the data.

Intended changes in document: Figures will be redrawn. Text will be clarified.

6. The opposite evolution in East Anglia and SW England might be correct (p11)? You need to look at the observations to check this. There is an outof-phase correlation between SE England and NW Scotland.

Author reply: We are not quite sure how NW Scotland relates to our analyses, but we have checked with the observationally derived series. We will further discuss this point in the new version. A comparison between more data series will be included, but it doesn't change the point here.

Intended changes in document: We will extend on the discussion.

7. If series (p12) have the order of one degree of freedom, then what are you doing showing them.

Author reply: The new version likely will not include this analysis.

Intended changes in document: The correlation analyses will be mostly or even completely absent from a new version.

Parts of this page are very difficult to read and follow.

Author reply: We will try to clarify this.

Intended changes in document: We will rewrite the results section.

8. No seasons are given with Figures 4 and 5.

Author reply: We will clarify the seasons throughout the manuscript. It is MAMJJ except when we additionally use JJA.

Intended changes in document: Captions will be clarified.

9. Trouet et al (2018) would have done better to have used the 300-year long instrumental records from the British Isles instead of going straight to tree-ring reconstructions. There are large variations across the British Isles with the size of the influence of the westerlies on precipitation amounts in the spring and summer. For example (p16) the NAO has no influence on East Anglian precipitation amounts in the winter half year. The NAO effect is much stronger on the western and northern areas of Britain, and it is mainly in the winter season. When you talk about spring/summer and the NAO are

talking about the same NAO as in winter? It would be useful to discuss how the North Atlantic Jet that Trouet et al (2018) talks about relates to the NAO, if it does?

Author reply: We will discuss the large scale circulation influence more extensively.

Intended changes in document: We will extend on the discussions of the large scale circulation.

10. P17 states that standardization of precipitation goes beyond comparing means and deviations. I'm not sure that you have shown anything other than just the means and SDs.

Author reply: Obviously, we disagree. We will clarify this point in the manuscript. Our analyses allows to compare the full distribution including measures that cannot be evaluated using the mean and the SD like the asymmetry of the distribution and its tails.

Intended changes in document: We will try to clarify the benefits of the SPI.

References Kendon M and D Hollis, 2014: How are UK rainfall-anomaly statistics calculated and does it matter? Weather 69, 37-39.

**0.2 Referee 2:**

While I think that this paper has merit and could provide interesting insight it is my view that it is not yet ready for publication. I encourage the authors to rethink the structure and layout of the paper and the key messages to be delivered. I think that such a paper would be welcomed by the field and of interest to the readers of the journal. But to reach a standard for publication significant work remains.

Author reply: Thank you. We are going to restructure the manuscript completely and clarify the key messages.

Intended changes in document: We will restructure the manuscript to provide more focussed information on our main points.

From the outset the specific aims of the paper are rather vague; the introduction section needs clearer structure. At the moment it jumps from one topic to the next without really unpacking where the state of knowledge it at in any aspect. The authors need to structure the introduction much more clearly, building the necessary context for the reader to understand what the aims are and the summary of information necessary to move to the next stage.

Author reply: We will more clearly structure the manuscript.

Intended changes in document: Introduction and also subsequent sections will be more clearly formulated.

If the focus is on the British Isles why just use the EWP series and not the Island of Ireland monthly series from 1711 or the Scottish regional series. I realise the latter is shorter, but to talk of the British Isles and not use the other available regional series is confusing. Murphy et al. (2018) cited in the introduction show that CET is also strongly correlated, at least at decadal scales with the Irish series.

Author reply: The focus is a small domain on the British Isles, not the whole of the

Isles.

Intended changes in document: We will clarify our spatial focus.

Why did the authors choose these tree ring reconstructions? To the best of my knowledge these are based on ring width reconstructions which have been shown to be less reliable for precipitation. Why not incorporate the oxygen isotope reconstructions done by Rinne et al. (2013) for southern England. Indeed in their discussion, if i recall correctly, they identify interesting points of departure from both EWP and Kew precipitation series for the summer months. Again in providing this suggestion as I am reading it is not clear what the time focus is of the paper – spring/early summer, spring?? The study design needs clearer thought, signposting and explanation.

Author reply: We will clarify the seasonal focus of the manuscript and the additional points you raise. We will argue for not using Rinne et al. in this context. Among other reasons: to our knowledge the data from Rinne et al. is not publicly available. The focus of the manuscript is an extended spring season. We will ensure that this becomes clear everywhere.

Intended changes in document: We will clarify our scope.

Regarding the selection of ensemble members from model reconstructions, why not use the entire ensemble? In the next paragraph it is noted that the selection is rather arbitrary and it is assumed that the domain sufficiently represents EWP domain. Some kind of table to help the reader interpret the different forcings used would be helpful.

Author reply: We are not quite sure what the reviewer is referring to but we will try to clarify this. We agree that the selection of the domain within a simulation is in a way arbitrary. We are going to move the analyses of the PMIP3-ensemble to a purely supplemental manuscript asset.

Intended changes in document: As far as this point is still relevant to a rewritten manuscript, we will try to clarify this. Analyses of the PMIP3-ensemble will be removed

from the main body of work.

The use of the SPI to investigate the 6.7 and 93.3 percentiles is a very stringent test of models and reconstructions is it not? The EWP is essentially a composite series and extremes are likely smoothed out. Also, is it a fair ask to expect climate model reconstructions to be able to represent these, especially if not employing a large ensemble? I am only asking out of curiosity here and would like to be informed of how stringent the comparison you are setting up is.

Author reply: We will discuss why we think the comparison of the distributions makes sense even for area average or composite series. By using distributions we essentially compare climate states which in theory should account for a part of the internal variability. Thus, assuming there is a common signal the evolution of the climatological properties could agree between data sets even without employing a large ensemble.

Intended changes in document: We are going to clarify the limitations and the stringency of our proposed method.

Any bias correction applied to the models? Does SPI negate this?

Author reply: No. We don't use statistically downscaled data. Bias-correction is not the scope.

Intended changes in document: We will clarify this.

Results presented in the methods section need to be moved.

Author reply: Will be changed.

Intended changes in document: Results from the methods section will be moved to later sections.

The paper is badly let down by plots that are very hard to decipher and methods applied that are not appropriately, or sometimes not at all, explained in the methods section.

Author reply: We will pay attention to clarify the visual presentation and the methods used.

Intended changes in document: We will take care to describe all methods and to present the results more clearly.

Fig 1 – no detail of the types of smoothing applied covered in the methods. What is a 'first impression' , not a scientific term. What CET time step is the smoothing applied to? Monthly or annual series. Why not plot as an ensemble rather than 11 sub plots? Line types in legend do not match the plots. Use of sunspot data is not covered in the data section so far as I recall.

Author reply: We will try to remedy all these points. We use a 51 point Hamming filter. We use the extended spring data here as well. We decided to use the 11 sub-plots since we regarded the ensemble plot to be even less visually helpful.

Intended changes in document: This Figure or a similar representation is likely going to be moved to a purely supplementary manuscript asset. Discussions of the sunspots will either be added or the data will be completely removed from the manuscript.

A table detailing the various data sources compiled is badly needed.

Author reply: We will present the used data in a clearer manner.

Intended changes in document: We will present the used data in a clearer manner.

The use of differing periods is confusing, how can this be comparative – which is the primary aim of the paper.

Author reply: We will try to be more clear in our thinking on how to compare the used data sets. However, we are unsure to what part of the manuscript this comments precisely relates.

Intended changes in document: The methods section will give more details on the comparisons.

Please think about presenting results in a clearer way. I literally spent hours trying to figure out what the figures were showing and in many aspects am no clearer.

Author reply: Thank you for spending so much time on the manuscript. We aim to clarify the presentation including a clearer outline of the spatial, methodological and dynamical considerations of the manuscript.

Intended changes in document: The new methods section will clearer spell out what is shown later, and Figures will be optimised.

There needs to be a more systematic approach to this work in terms of presentation and some sub sectioning in the results and discussion to help the reader.

Author reply: We will try to lead the reader more clearly through our thinking.

Intended changes in document: We will try to structure the results-section more clearly and to more systematically direct the reader through the manuscript.

The title of the paper concerns precipitation. It is confusing to start the results off with temperature.

Author reply: Discussions of temperature will be minimised in a new version and not start the results.

Intended changes in document: The results section will be restructured.

I find it next to impossible to interpret the caption of Figure 2.

Author reply: We will clarify the presentation of Figures and captions.

Intended changes in document: Figure captions will be clarified.

It is difficult to comment in much depth on the nature of the results and the points made in discussion and conclusion given how difficult it is to decipher what was done.

Authors need to revise the structure of the paper to systematically consider the inconsistencies of interest.

Author reply: We will try to make our points more clearly in a new version.

Intended changes in document: A rewrite of the manuscript is necessary.

**0.3 Referee 3:**

Summary: The manuscript involves a comparison of climate model simulations with an ensemble of global and one regional model to long observationally-based records and two paleoclimate reconstructions. Little consistency is found between time histories of these records, suggestive of a large role for internal atmosphere-ocean variability. Importantly, while there is little agreement between the characteristics of the model simulations and the observationally-based records, these differences do not appear to be systematic across models and cannot be explicitly linked to model bias. Likewise, there appears to be even less agreement between the characteristics of the observationally based records and the reconstructions. Together this work is consistent with mounting evidence that regional hydroclimate is largely "unforced".

General Remarks: While the manuscript is interesting and highlights some important results, it is at times unclear what should be taken away from the results. This is, in part, an issue with the introduction and a refocused introduction that clearly describes the motivations and goals of the study would greatly improve the manuscript. Below are a number of specific and more general comments.

Author reply: A new version will clarify the introduction not only with respect to the motivation and the conclusions but also related to the general focus and intention and peculiarities of our approach in comparing different sources of information.

Intended changes in document: The complete manuscript is going to be restructured.

Page 1, Line 8: and in the standard deviations seems a weird statement.

Author reply: To be changed.

Intended changes in document: We will modify the abstract.

Page 1, Line 18: add "of" before "whether".

Author reply: Will be changed.

Intended changes in document: Will be changed

Page 1, Line 19: what is meant by requires consistency?

Author reply: We will clarify our idea of requiring consistency.

Intended changes in document: Introduction and discussions are going to be more explicit about what we mean by consistency.

Page 1, Line 21: suggest changing to "over approximately the last 350 years".

Author reply: Will be modified.

Intended changes in document: Will be modified.

Page 2, Line 2: suggest removing "in particular".

Author reply: We will rephrase the sentence.

Intended changes in document: Will be rephrased.

Page 2, Line 6: change "base" to "basis".

Author reply: Will be changed.

Intended changes in document: We will change the sentence.

Page 2, Line 10: change "compare directly" to "directly compare".

Author reply: Will be changed.

Intended changes in document: Will be changed.

Page 2, Line 12: Cooper and Wilson et al. are the reconstructions. I would be careful here and throughout with the semantics of "data".

Author reply: We will try to be clear in how we refer to the various

sources of information. However regarding the longstanding discussions on what may be named "data", the Wiktionary writes, slightly paraphrased, at https://en.wiktionary.org/wiki/dataEnglish: data: a) Information, especially in a scientific or computational context, or with the implication that it is organized. b) Recorded observations that are usually presented in a structured format. c) A representation of facts or ideas in a formalized manner capable of being communicated or manipulated by some process. Data in the context of our writing is generally any set of information.

Intended changes in document: We will carefully consider how we describe the various sets of information.

Page 2, Line 16: You argued in the paragraph above that you do not want to use gridded reconstructions. I understand that this paragraph is addressing a new issue but the reference to the OWDA thus seems unusual here. In general, this paragraph does not seem necessary. I might instead start at the beginning of the next paragraph and add a statement at the end of that first sentence saying that you are doing the standardization to make the reconstructions directly comparable to SPI.

Author reply: We will try to more clearly justify the choice of method and data.

Intended changes in document: We will rephrase the introduction to ensure a logic reading.

Page 2, Line 24: Suggest changing "their data" to "the utilized archives".

Author reply: We will rephrase the sentence.

Intended changes in document: Will be phrased differently.

General comment: A lot of the above reads much more like a methods section than an introduction. I suppose this is more of a personal preference but the paper might be more impactful with a standalone introduction that does not include this methodological information.

Author reply: We will try to better separate introduction and methods.

Intended changes in document: We will take care to clearly distinguish between introductory comments and the description of the methods and the data..

Page 2, Line 24: The sentence about Murphy et al. (2018) feels out of place. I would try to tie this into the paragraph above or remove it.

Author reply: We will try better to embed the point of Murphy et al. (2018).

Intended changes in document: Introduction will be rephrased.

Page 2, Line 28: Suggest removing "than in periods that are more recent".

Author reply: We will restructure the sentence.

Intended changes in document: We will phrase the sentence more clearly.

Page 2, Line 29: Suggest splitting the sentence after the Maunder Minimum dates. I would then reword as: "Instead, they generally start around the late 18th century, when sunspot numbers indicate a period of relatively strong solar activity (Clette et al., 2014), and thus also include the transition. . ."

Author reply: We will clarify the point.

Intended changes in document: The paragraph will be modified.

Page 2, Line 35: Suggest changing "in European subdomains" to "across Europe".

Author reply: We will modify the sentence.

Intended changes in document: We will make the point more clearly.

Page 3, Line 1: Change "extend" to "extent".

Author reply: Will be changed.

Intended changes in document: Will be changed.

Page 3, Line 10: This sentence is long and the second half I had trouble understanding. Perhaps you could split this up into two sentences and expand on the point that you are trying to make in the second half of the sentence.

Author reply: We will try to clarify the point.

Intended changes in document: The paragraph will be clarified.

Page 3, Line 20: Suggest "using the global model ECHO-G for boundary conditions" instead of "externally forced". I am also not sure what this part of the sentence means: "and reconstructions over larger regional domains."

Author reply: We will clarify the sentence.

Intended changes in document: We will make the point more clearly.

Page 4, Line 1: Suggest changing "and the simulation data representing" to "and simulations that often represent".

Author reply: We will adapt the sentence.

Intended changes in document: Will be changed

Page 4, Line 2: Suggest changing "evaluation" to "comparison".

Author reply: Will be changed

Intended changes in document: Will be changed.

Page 4, Line 17: Change "allows comparing" to "allows for the comparison of".

Author reply: Will be changed

Intended changes in document: Will be changed.

Page 4, Line 19: Change "allows evaluating and comparing" to "allows for the evaluation and comparison of".

Author reply: Will be changed.

Intended changes in document: We will rephrase the sentence.

Page 4, Line 22: Suggest changing "extends the available metric for assessing the agreement in" to "allows for the rigorous comparison of".

Author reply: We will try to discuss this point more clearly.

Intended changes in document: We will extend on this point in a new version.

Page 4, Line 23: Suggest changing "not only for periods without but also with" to "for periods both with and without".

Author reply: We are going to change the sentence.

Intended changes in document: Will be changed.

General comments on introduction: I am unsure about the relevance of short-term (decadal) relationships between temperature and precipitation with those expected as a result of climate change (first two sentences of the introduction). The relationship between hydroclimate and temperature at the end of the 21st century in climate models is largely due to evaporative demand, which has a first order impact on water storage but not necessarily on precipitation. These changes are also very large in magnitude, and co-occurring with large magnitude changes in plant physiology, making deeper-time paleoclimate comparisons more appropriate for evaluating climate models (e.g., Scheff et al., J. Clim., 2016). I do not think this precludes such analyses being useful, I am just unsure of using the relationship between temperature and precipitation with an eye towards climate change as the motivation.

Author reply: We are going to adapt the motivation to address this point and to provide a more focussed impetus for our study. The link between temperature and precipitation is more complex, and not only restricted to long time scales. It may be modulated even at interannual timescales by other processes, for instance, through the link between

temperature and cloud cloud cover during the extended summer season. Thus, an analysis of the covariability between temperature and precipitation even at interannual and decacdal time scales serves as a validation of both reconstructed variables on the one hand, and of the corresponding link between these two variables in climate models on the other hand.

Intended changes in document: The motivation will be rephrased.

I would be careful with the semantics of the word data to make sure that things are as clear as possible. Likewise, I would refer to reconstructions, observations and simulations each with a single consistent term. This applies to the entire manuscript.

Author reply: We will try to be consistent in the descriptions of the various sources of information.

Intended changes in document: As mentioned above, we will take care to be clear in our use of the term data and its application to the various sources of information.

The introduction bounces around a lot, with quite a bit of methodology (see general comment above). I think that as cast it will leave the reader uncertain about the motivations and goals of the study. I suggest that the authors revisit the introduction with an eye towards clarity.

Author reply: We are going to try to motivate our study more clearly and to provide the reader with a better picture of from where we start and where we try to go.

Intended changes in document: The introduction will be reformulated.

I made an effort to make grammatical edits in the introduction but likely missed some. I will not be able to make this effort in subsequent sections but suggest that the authors revisit the manuscript with an eye towards grammar and syntax.

Author reply: We are going to try to improve the language, once more.

Intended changes in document: We are going to improve the language.

It might be worth explicitly outlining how what you are doing here is different from Gómez-Navarro et al. (2015). Along with what is described the methods there would appear to be quite a bit of overlap.

Author reply: To be clear, we are co-authors on GN15. GN15 look at a regional simulation and gridded reconstruction over the European domain. They, among other things, compare both for a variety of regional sub-domains and a number of different datasets. They do not consider the small regional scale, they do not consider the SPI, they always have the spatial reconstruction step.

Intended changes in document: We will clarify the difference between Gómez-Navarro et al. (2015) and our manuscript.

Page 4, Line 28: Change to "in the form".

Author reply: We will modify the sentence.

Intended changes in document: Will be modified.

Page 5, Line 15: Suggest adding "In particular," at the start of this sentence to link it to the previous sentence. Suggest also changing "different means" to "systematic differences in the values of".

Author reply: We will adapt the paragraph

Intended changes in document: We will change the paragraph.

Page 5, Line 16: Suggest "While model-biases may also contribute to these differences,. . ." and change "bias" to "source of differences".

Author reply: We are going to adapt the paragraph.

Intended changes in document: The paragraph will be clarified.

Page 5, Line 17: I doubt it matters but why the different domain here?

Author reply: The domains for CET and EWP differ, thus we also adapt different model

domains. However, a new version will have a smaller role for the temperature data.

Intended changes in document: We will remove much of the temperature discussion from the manuscript but also discuss the different domains more clearly if necessary.

Page 5, Line 30: Change "to include" to "the inclusion of".

Author reply: Will be changed.

Intended changes in document: To be changed.

Page 6, Line 5: Change "allows to compare" to "allows for the comparison of".

Author reply: Will be changed.

Intended changes in document: To be changed

Page 6, Line 19: Change "allows considering the changing amount of precipitation" to "allows for a robust quantification of changes in precipitation amounts between subsequent periods, for instance ".

Author reply: We will clarify this paragraph.

Intended changes in document: Will be clarified.

Page 7, Line 2: Remove "just".

Author reply: Will be changed.

Intended changes in document: To be changed.

Page 7, Line 3: Add "the" before "time series".

Author reply: Will be changed.

Intended changes in document: To be changed.

General comments on methods: The half-degree simulations are course resolution for a regional climate model. At least one of the last millennium simulations analyzed is

one degree (CCSM4), how much added information do we expect from a regional simulation at this course resolution and what physical processes is it capturing to provide that information?

Author reply: We will comment on this. See, for example, the most recent papers by Ludwig et al. (2018) and Sørland et al. (2018).

Intended changes in document: We will more clearly discuss the benefit of even a slight increase in resolution and why a regional simulation adds more benefits than just an increased resolution.

Page 7, Line 13: Change "tentative" to "qualitative".

Author reply: Will be changed.

Intended changes in document: Will be changed.

Page 7, Line 20: Suggest change the last sentence to "This is likely to also impact our analyses of precipitation".

Author reply: We may modify the sentence.

Intended changes in document: We will clarify this paragraph.

Page 7, Line 32: What is the European domain?

Author reply: We will detail the domain.

Intended changes in document: In case it is still relevant in a new version, we will be specific about this larger European domain.

Page 8, Line 2: Suggest removing the first sentence.

Author reply: We are going to restructure the description of our results

Intended changes in document: The results section is going to be rephrased and re-structured.

Page 8, Line 5: Suggest changing "representations" to "time series".

Author reply: Will be changed.

Intended changes in document: To be changed.

Page 8, Line 7: Suggest removing "but the Southern-Central England data enters it later".

Author reply: We will clarify the description of the results.

Intended changes in document: The results will be clarified.

Figure 2, caption: Why call the Southern-Central England record SW England in the legend?

Author reply: Will be changed. Thank you for pointing out this oversight.

Intended changes in document: To be changed.

Page 10, Line 23: Change "allows evaluating" to "allows for the evaluation of".

Author reply: We will clarify the sentence.

Intended changes in document: This will be clarified.

Page 10, Line 24: Change "gliding" to "sliding".

Author reply: Will be changed.

Intended changes in document: To be changed.

Page 10, Line 25: Suggest removing "partially".

Author reply: Will be removed.

Intended changes in document: To be removed.

Page 10, Line 27: Change sentence to read "The moving window transformations show

the percentiles represented by a given amount of precipitation over time (Figure 3)."

Author reply: We are going to clarify the procedure.

Intended changes in document: This part of the manuscript will be clarified.

Page 12, Line 1: Suggest changing "We pointed above at" to "In the previous sections we described".

Author reply: We are going to modify the sentence in question.

Intended changes in document: This part will be clarified.

Page 12, Line 6: Suggest changing "gliding" to "sliding".

Author reply: Will be changed.

Intended changes in document: To be changed.

Page 12, Line 11: Suggest combining these two sentences.

Author reply: We are going to make the point more clearly.

Intended changes in document: We are going to change this paragraph.

Page 12, Line 12: Suggest changing "Considering" to "In".

Author reply: We are going to modify the sentence.

Intended changes in document: To be changed.

Page 12, Line 15: Suggest removing "correlation".

Author reply: Will be removed.

Intended changes in document: To be removed

Page 12, Line 20: Suggest changing "highly" to "strongly".

Author reply: Will be changed.

Intended changes in document: Will be changed.

Page 12, Line 21: Why "CET" here and not elsewhere?

Author reply: We are going to be more consistent in the use of abbreviations.

Intended changes in document: The new version will be consistent in use or avoidance of abbreviations.

Page 12, Line 31: Change "very low frequent variability" to "low frequency variability".

Author reply: Will be changed.

Intended changes in document: To be changed.

Page 15, Line 8: Again why the use of "CET" here and not elsewhere?

Author reply: We are going to be more consistent in the use of abbreviations.

Intended changes in document: A new version is going to be consistent in use or absence of CET, EWP, and other abbreviations.

Page 15, Line 15: Why just atmospheric circulation when coupled variability can also do this?

Author reply: Indeed. We will change this and discuss more extensively factors influencing the regional domain.

Intended changes in document: We will clarify this discussion.

Page 15, Line 24: I found this paragraph difficult to understand. The final sentence is seemingly important but I was unclear on what it means. Likewise, I would clarify what is meant by unfortunate earlier in the paragraph.

Author reply: We are going to clarify our thinking on regional climate variability, natural forcing, the relation between temperature and precipitation, and the precipitation distributions.

Intended changes in document: A new version will discuss this more clearly.

Page 16, Line 7: Suggest changing "appears" to "is".

Author reply: Will be changed

Intended changes in document: Will be changed.

Page 16, Line 23: While this is true, it is unclear how it relates to the other discussion.

Author reply: We are going to better connect the discussion on changing teleconnections to the discussions on internal variability and the representativeness of data sources.

Intended changes in document: Discussions of a new version will be more clear in this discussion.

Page 17, Line 19L Change "source" to "sources"

Author reply: Will be changed

Intended changes in document: To be changed.

A note is in place on potential added references. There are a number of topics, which may need discussing additional references.

First, there is the SPI. We have not yet decided which of the previous studies using the SPI in paleoclimatology are essential for our argumentation. Candidates include Domínguez-Castro et al. (2008, doi:10.1016/j.gloplacha.2008.06.002) Machado et al. (2011, doi:10.1016/j.jaridenv.2011.02.002), the SPI use by Lehner et al. (2012, see original references), Seftigen et al. (2013, doi:10.1002/joc.3592), Yadav et al. (2015, doi:10.1016/j.quascirev.2015.04.003), and Tejedor et al. (2016, doi:10.1007/s00484-015-1033-7). These, however, mainly deal with the SPI as original reconstruction target.

Second, as we noted above, we have to discuss why we think the used regional climate model has indeed a chance to improve on the representation compared to the PMIP3-ensemble. Recent publications by Ludwig et al. (2018, doi:10.1111/nyas.13865, Sørland et al. (2018, doi:10.1088/1748-9326/aacc77), and Pinto et al. (2018, doi:10.1002/joc.5666) allow to make this point. These may also become relevant in discussing how our work differs from Gómez-Navarro et al. (2015).

Third, while we in principle think that our references for contextualising regional climate variability and the large scale are sufficient, we may include additional discussions on the relation between the large scale climate dynamics and precipitation (e.g., Jones et al., 1993; Mayes, 1996; Wilby et al., 1997; Osborn and Jones, 2000; Murphy and Washington, 2001; Wedgbrow et al., 2002; Kingston et al., 2006; Lavers et al., 2010; ).

Fourth, there remains the question, how much of the literature on the British observational datasets is relevant to the discussions. Our initial assessment was that the main references for the datasets are enough. Possibly, additional references will be added (e.g., Wigley and Jones, 1987; Gregory et al., 1991; Jones and Conway, 1997; Kilsby et al., 1998; Osborn et al., 2000; Croxton et al., 2006; Marsh et al., 2007; Simpson and Jones, 2012; Simpson and Jones, 2014).

It remains to be seen to what extent discussing the issues of the used and not considered reconstructions requires additional references.

Once more, thank you for your help.

On behalf of the authors

Yours sincerely

Oliver Bothe

List of relevant changes to the manuscript:

Overall manuscript:
- restructured
- title slightly changed

Abstract:
- partially rewritten

Introduction:
- restructured
- partially rewritten
- Discussions on uncertainty of comparison studies extended
- Discussions on SPI extended
- Discussions on all types of data extended, i.e. on simulations, reconstructions, and observations
- Discussions on consistency extended
- Discussions on our expectations extended

Data:
- rewritten
- data table added
- Discussion of choice of parameter, domain, data-types, and data-sources extended

Methods:
- rewritten
- Discussion of SPI extended
- Discussion of Smoothing added

Results:
- Figures redone
- PMIP3 removed to supplementary asset
- Plot of precipitation added
- more instrumental data included
- observational indices for subdivisions of the England-Wales precipitation newly included
- Correlation analyses added
- rewritten
- relation between temperature and precipitation minimised

Discussions:
- restructured
- rewritten
- Discussion of SPI extended
- Discussion of data extended
- Discussion of additional data added

- Discussion of approach extended
- Discussion of results extended
- Discussion of additional results added
- Discussion on internal variability extended
- Discussion on dynamics extended

Conclusions:
- slightly rewritten

Appendices:
- partially rewritten
- Distributional parameter plots added

Supplement:
- added
- additional Figures
- additional analyses

[revised manuscript text omitted]

**4.1.2 (Paleo-)observational data and regional simulation output**

The observed Central England Temperature (CET)is Figure 3 presents the two reconstructions and the only data whose England-Wales precipitation in comparison to the respective data from the regional simulation. All data are again for the extended spring season from March to July (MAMJJ), and the panels zoom in on the period of the regional simulation. We show the interannual time series and the 51-point Hamming filtered series shows some agreement to changes of the decadally averaged sunspot numbers. CET starts from a cold period prior to 1700 and then reaches a plateau of higher temperature that is intersected by short cold episodes around 1750 and early in the 19th century.

The regional simulation similarly has cold conditions about 1700 and then warms until the early second half of the 18th century with a subsequent transition to cold conditions in the Hamming-filtered representation.

[Figure]

a) East Anglia, reconstruction & model

b) Southern–Central England, reconstruction & model

c) England–Wales, observations & model

——— MAMJJ data ——— MAMJJ model data

**Figure 3.**  Extended spring (MAMJJ) precipitation in light grey in background, divided by 100). Light colors are European domain large-scale mean temperatures.observational CET, bCCLMsimulation, cMRI, dMPI, eIPSL, fHadCM3, gGISS-E2-R 21, hGISS-E2-R 24, iGISS-E2-R 27, jCSIRO, kCCSM4.Vertical lines give the years 1700, 1784, and 1816.~~

[revised manuscript text omitted]

---

## Author Response (AR2)

Dear editor, dear referees,

We would like to thank the reviewers and the editor once more for their time spent on reviewing our manuscript and their comprehensive comments and ratings.

In the following we reply (normal font color) to the comments (red) and note implemented changes.

Best Regards

On behalf of the authors

Oliver Bothe

**Editor**

dear authors

I now got two reviews back on your revised version. The reviewers acknowledge the effort that have been made and they appreciate the work you have done for the revisions which have improved the workd in various aspects. Both have some additional comments/suggestions how the paper can be improved. I think both reviewers provide clear instructions for improvements. Can i therefore ask you to follow the reviewers comments and revise your manuscript accordingly?

with my best wishes and looking much forward to the new version

Jürg

**Response to reviewer #1**
**Referee 1**

Second Review of Bothe et al.

The authors have improved the paper in this iteration. Some further work remains to be done however.

First we would like to thank the reviewer for their comments.

Then, we would like to express a general comment. We read in the reviewer's comments a fundamental scepticism concerning all data sets that we use in our analyses as well as the analyses themselves.

Obviously, each of the data sets has its associated uncertainties and deficiencies. This is valid for observations, reconstructions, and model simulations. We explore here to what

extent common variations can be expected and to what extent they are present in the used data sets. We do so not only for the time-series but also for further properties of the distributions of the precipitation data sets.

Our revisions try to consider the reviewer's comments as far as they are within the scope of this manuscript. We explain our reasons when we do not integrate all reviewer's suggestions.

Please see the document including tracked changes, for our edits.

The paper is longer than it needs to be, especially the discussion.

In the revised version, the discussions were notably shortened.

Following the suggestions of the reviewers we reconsidered all new results and moved part of these to the results section (see new section 4.5).

We moved all discussions of data sets and methods to the data and methods sections (see sections 2 and 3 and particularly the tracked changes) and tried to shorten these parts. We did not necessarily succeed as we see much of this as required by the previous round of revisions.

There is also work to be done on teasing out the key contributions. At least the paper should draw out where there is agreement and not between sources, ie. Provide the key periods. This would be usefully done in a table and would help focus future research to further understand the issues raised.

We do not provide a table but we try to clarify these points. Agreement and disagreement is more clearly highlighted throughout the results section (see section 4). The key points and key periods are further highlighted in the discussions (see section 5, and particularly section 5.2) and conclusions (see section 6).

The conclusions now try to clearly point to all the contributions of our work (see tracked changes in section 6).

The integration of tree rings (based on width) is done so in the full knowledge that previous work by the authors of these data sets have shown that there is a lack of stability in terms of the relationship with station based precipitation. This work confirms a well established fact, that there is inconsistency between tree ring reconstructions and station based observations. I am therefore not sure what this aspect adds. This is well known. Surely this paragraph from the discussion negates the use of the tree ring series you use for the purpose you use them for.

"Young et al. (2015) conclude that these differences make it unlikely that the tree-ring based works and their δ 18O based work represent the same environmental parameter, and they emphasize the lack of a calibration against regional precipitation data. They further discuss

the reasons given by Wilson et al. (2013) for the lacking stability of the Wilson et al. reconstruction, namely, different climate-sensitivity of the trees, unreliable instrumental data, and pollution. Young et al. (2015) conclude that their own data reliably reflects precipitation while the tree-ring widths most likely represent the combination of various environmental influences on tree growth instead of a single climate parameter."

All data sets, i.e. reconstructions, models and observations, are uncertain. However, all data sets also contain some part of the information we are interested to tease out in paleoclimatology. We openly present these uncertainties.

If we take the logical consequence of the reviewer's somewhat radical interpretation and additionally consider that our supplementary analyses show comparable weaknesses in the isotope based records, then there are no reconstructions left to compare to until we obtain an isotope enabled regional simulation which we could directly compare to the isotope measurements.

Our analyses try to identify to what extent the different data sets agree or disagree in view of their respective uncertainties, and we discuss the possible reasons for these agreements and disagreements. As such, we are convinced it is a valuable contribution.

Our aims are a) "to study the consistency in the statistical properties of precipitation distributions in these sources of information" and b) to do so by using the SPI and "[motivating] the use of the Standardized Precipitation Index in hydroclimatic comparisons between different data sets in paleoclimatology". These sources of information here includes all available types of data, observations, reconstructions, and simulations.

We are convinced that the known weaknesses of the reconstructions do not negate either of these goals.

Climate models need to be tested at long multi-centennial timescales relevant for future climate change. At these long timescales, the available evidence from proxies and long instrumental records becomes more uncertain. Nevertheless, we think that our study is a valuable contribution towards that goal.

To help tease out inconsistencies I am not sure why you don't use series such as the Paris London Index, London Sea Level Pressure (both by Richard Cornes) or perhaps the Westerly Index to help point to which data sources that potential errors derive from. There is more data available to help this work. It seems a missed opportunity not to use them.

Indeed, it is of value to study every available data set giving information about British Isle climate over the last centuries - in the case mentioned by the reviewer since 1692. Our focus here is precipitation. Detailed analyses on the involved dynamics are beyond the scope of our manuscript.

This is not so much a decision on principle but the judgement that the various sources of information available (proxy-based index reconstructions, observation based indices, proxyand observation-based field reconstructions, data assimilation efforts, and [regional] simulations) require a separate manuscript to assess them against each other. Indeed, Figures 11 and 12 of Cornes et al. (2013) indicate on the one hand large discrepancies between different estimates of North Atlantic Oscillation indices but on the other hand reasonable to strong common variations. That is, teasing out reasons for disagreement in precipitation estimates is only possible after teasing out reasons for the disagreement in circulation estimates.

More generally, using pressure related indices does not a priori provide evidence on the reasons for uncertainties and disagreement in and among the data sources we use.

Also, regarding the data, EWP is a composite drawing on stations as far north as Edinburgh. Yet the analysis, reconstructions and modelling is based on southern England. It is not surprising to me that inconsistencies exist given the study design.

We would like to point to a number of issues here.

First, the modelling is not solely based on southern England, we select only data from these domains from our simulation output covering the entire European realm.

Following the suggestions in the first round of reviews, we included a number of additional data sets. Using these data sets, we could see different levels of agreement among the observational or observation based data sets. These comparisons further show that the reconstructions compare worse to the local to regional observational data sets than the observational data sets compare among themselves (e.g., Figures 1 and 2, and Supplementary Figures 1 to 3).

In our understanding, this indicates that the inconsistencies are not due to the uncertainties in the data sets in general and not due to a biased choice of data sets in our study.

Simulation results are based on a single RCM model chain. Can this provide a robust estimate for comparison with observations and reconstructions? You have not convinced me.

To our knowledge, there are only two regional simulations for this area for the last few centuries. Both simulations are not totally independent, since both have been driven by different versions of the same global model from the Max-Planck-Institute.

Using an ensemble would of course make the results more robust. However, such an ensemble does not exist. Regional simulations are costly, and only few modelling groups run regional paleoclimate simulations.

As in the case of a single observational record or of a reconstruction without uncertainty estimates, the comparison cannot be complete. However, we are confident that there is an added value in such comparisons. Single simulations are often the only available spatially complete physically consistent information about past climates. Evaluating how they

compare to reconstructions and observations is in itself of value for our understanding of past changes. We do this for this simulation, the reconstructions, and the observations in our chosen study region.

Any single simulation has to be evaluated. We do this for this simulation in this region.

The discussion is far too long and contains some repetition, making it difficult to read. It needs to be considerably shortened and sharpened in its focus on key points. It needs to distil the key contributions of this work. The structure is also odd, there are concluding remarks followed by conclusions, and section 5.3 is Discussion of Results? The way it is organised re-traces the entire paper again. It needs to get to the job it is supposed to do, i.e. discuss key points from the findings.

We shortened the discussion and tried to focus it more. The concluding remarks for the discussions were removed. Discussions of data and methods were moved from section 5 to sections 2 and 3 respectively. The additional results were moved to section 4.5 and shortened. The discussions, thus, solely focus on the discussion of our findings, their validity, and their implications (see section 5).

The conclusions in section 6 try to highlight our main points.

It is my opinion that you give too great a credit to the early observations. These are likely to be highly uncertain. Indeed for many of the series you use the early parts are composites taken from weather diaries and very early instrumentation. In addition do you use the corrected Oxford series of Burt and Howden who corrected for gauge being on a roof and changes in gauge design? Or is it the original series? You could look more critically upon the observations.

As the reviewer may see from our section on the observations and our table, the observational series stem either from the Met Office homepage or are from the Global Historical Climatology Network (GHCN) in the version as provided by the Climate Explorer. For the GHCN, please see Peterson and Vose (1997).

We already acknowledged that the observations also have large uncertainties in the previous version. This study is a comparative analysis of the available information - observations, proxies, and modelling. We provide possible explanations for the lack of agreement among these data sets when discussing the data sets in section 2 and in section 5. We cannot revisit all single time series from publically available compilations.

We do not further address the uncertainty of the observations beyond what is already in the paper.

Given that we know the reconstructions have issues, it would be useful to highlight to the reader the particular periods where we have agreement between the different sources and those that we do not. This might help focus and provide a priority for future work. What I am saying is that the results need to be consolidated and key insights distilled. What you are

saying at the moment is that there is disagreement between the observations and reconstructions, disagreement between both of these and model simulations but the latter is not as different to the observations as the reconstructions are. I need more insight on this - are there periods where all agree/disagree etc.

We try to clarify this and to highlight the disagreement and the agreement throughout sections 4 to 6. Indeed, already the last version included many pointers to the specific disagreements and agreements in the text of section 4. This allows any interested reader to find periods for future research foci. We try to extend these pointers where it appeared necessary.

We note that given the heterogeneity in the patterns of agreement and disagreement between the different sources it might appear overconfident on our side to point to specific periods for showing an exceptional high degree in consistency.

Please note that the negative relationship between temperature and precipitation does not include winter where the relationship is warm and wet. The way it is stated in the text gives the sense that it is year round. It is my understanding that temperature/precipitation relationships are strongest in winter and summer and much more variable in spring, which is why the signal may be variant in your comparisons.

The manuscript stated clearly that we are only considering the extended spring season. Crhová and Holtanová show that the negative relation also holds in spring in the observations.

We tried to reduce any chance of misunderstandings in all sections.

Again, this is not helped by the choice of your study season, which is different from the tree ring season and makes the entire analysis hard to follow at times.

We are unsure what the reviewer refers to with this comment. The season we study is the season used in the publications of the tree ring-width-based reconstructions. It is always MAMJJ except when we discuss other reconstructions that were included after the suggestions in the previous round of reviews.

We tried to be more precise in the seasonal attribution of our writing to reduce the amount of misunderstandings in all sections.

Is there a mistake in stating that Rinne et al calibrated to Oxford? My memory is that it was Kew they used. I may be wrong but please double check.

Rinne et al., 2013, page 16: "Finally, for the final reconstruction of precipitation, total May--August precipitation values were obtained by scaling the Woburn chronology against the Radcliffe precipitation series. i.e. the mean and variance of the chronology were set equal to those of the instrumental series over the calibration period 1815--1893."

Rinne et al., 2013, page 24: "The correlation coefficient between the reconstruction and the instrumental precipitation record from the Radcliffe station over the calibration and verification period exceeds the tentative minimum value required (r = 0.71) to provide acceptable results for palaeoclimatological studies (McCarroll and Pawellek, 2001). The precipitation series, which was formed by combining reconstructed values (1613--1893) and instrumental data (1894--2003), indicates significant decadal and centennial precipitation variability culminating in dry conditions in the early-middle 17th century and the late 20th century."

A key objective of the paper is to show the value of the SPI over moving windows allows a better comparison of datasets. You need to spell out this contribution more, be explicit. I have not been convinced of its merits above other approaches from this paper. For instance, in the abstract you state that SPI is a valuable tool for bridging part of the problem in assessing agreement and disagreement – what are the problems it addressed.

Simulations, reconstructions, and observations refer to different spatial scales and show different skillful scales. Therefore, we do not expect agreement between them in mean or variations a priori, especially for variables with a very localized and complex nature like precipitation.

The SPI effectively leads to the comparison of climatologies over moving windows. It allows to filter out higher frequency variations and allows to compare moments and percentiles of climatologies. It allows addressing general tendencies towards dryness or wetness relative to the location.

We do not state that the SPI is outperforming other approaches in any aspect but that it extends and supplements the other approaches.

I think that this is an important part of the paper as mentioned above we know there is disagreement between reconstructions and observations – what new does the SPI analysis bring, what further insight?

We try to clarify the insights in the variations of the percentiles and of dryness and wetness, which are easily obtained by the SPI. That is, among other things, the SPI provides information on the consistency of percentiles, moments and how these change over time and, thus, allows a deeper, more informative coherent comparison of hydrologic variables between different data sets.

Again in the abstract, what do you mean by the most recent historical period. This is not defined.

We thank the reviewer for pointing this out. We clarified this.

The table on date sources used needs to include the years covered by each

This is now included.

**Response to reviewer #2**
Referee 2

General Remarks: This is an update to a previously reviewed manuscript. The writing and clarity are significantly improved, however, there are a number of changes that must be made before publication. Please see the specific comments below.

We thank the reviewer for their valuable comments and suggestions that were mostly implemented as suggested. For a detailed response please see our replies below and compare the document with the tracked changes.

The one more general comment that must be addressed is that the discussion and conclusion sections need to be more clearly written. In particular, there are parts of these sections that read like methodology and results. These parts should be replaced with clear and concise statements on what can and cannot be taken away from the results—what do the results tell us and why is that important?

The discussion and conclusion sections have been shortened. Parts considering data and methods have been moved to sections 2 and 3 and shortened. Discussion of results has been moved to section 4.5. Paragraphs containing introductory statements or summaries of the results have also been moved or removed. The conclusions are now listed in a hopefully clearer manner (compare section 6).

Page 1, Line 1: Suggest changing "The Standardized Precipitation Index is a valuable tool for bridging part of the problems in assessing agreement and disagreement between the different sources of information. We assess the agreement in the temporal evolution of percentiles of the precipitation distributions" to "The Standardized Precipitation Index is a valuable tool for assessing agreement between the different sources of information, as it allows for a comparison of the temporal evolution of percentiles of the precipitation distributions".

Done

Page 1, Line 8: Suggest changing "consistent relations" to "consistency".

Done

Page 1, Line 9: Suggest changing "purely" to "impact of".

Done

Page 1, Line 9: I do not understand what is meant by the sentence that starts on this line.

We change the sentence as follows: The disagreement between sources of information reduces our confidence in inferences about the origins of hydroclimate variability for small regions.

Page 1, Line 22: What is meant by "supports", can you be more specific here?

We add: It strengthens our confidence in inferences from the consistent sources of information.

Page 2, Line 3: Suggest removing "and inconsistency".

Done

Page 2, Line 21: Suggest removing "complementing the current suite of statistical diagnostics for data-model comparisons".

Done

Page 3, Line 25: Suggest removing "regional instrumental series".

While we still think it is important to stress this difference to the previous work, we remove this.

Page 3, Line 28: Add "at" between "looking how".

Done

Page 4, Line 4: I have never heard the term "document asset". My understanding is that Climate of the Past uses the term supplementary material, although that may have changed. This note is just to make sure to use the correct term throughout.

To ensure stability of our supplements, we aim at providing them via https://osf.io/duyqe/ instead of uploading them to the Climate of the Past homepage. Compare https://publications.copernicus.org/for_authors/manuscript_preparation.html:

"Authors have the opportunity to submit supplementary material with their manuscript, such as additional figures and tables, highly detailed and specific technical information, user manuals, maps, or very large images. Supplements to articles should not be used as an archive for data. Therefore, data sets, movies, animations, or computer programme code should be deposited in FAIR-aligned data repositories and to insert a persistent identifier, ideally a DOI, in the manuscript."

Therefore we chose the word asset. We have modified this now.

Page 4, Line 7: Change "most" to "mostly".

Done

Done

Page 5, Line 26: Suggest adding "records" after "temperature".

Done

Page 6, Line 9: Suggest adding "observations" after "precipitation".

Done

Done

Done

Changed to: high-resolution modelling

Done

We add: The choice of 40 data points is an ad hoc decision that lies between the recommendation by McKee et al. (1993) of 30 samples and our window length of 51 years.

Done

Changed as suggested

Page 9, Line 14: Some background on what the Weibull standard deviation can tell us would be useful here.

To provide more information about the precipitation distributions, we choose to provide the square root of the second moment (the variance) to visualize our results. We add: this provides an additional clarification of how the precipitation distribution changes over time.

Page 9, Line 26: Change "Figures" to "figures".

Done

Page 11, Line 4: Suggest adding "sections" after "following".

Done

Page 12, Line 14: Why the choice of non-overlapping 11-year averages?

We add at the end of the paragraph: We choose 11-year non-overlapping windows to balance the number of available data points and the filtering of interannual variability.

Page 12, Line 16: Suggest changing "gives also" to "also gives".

Done

Page 14, Line 2-3: Can you quantify this?

We add: e.g., interannual pairwise correlation coefficients are about 0.9 between the simulated East Anglia data and the other two regions, while the simulated England-Wales precipitation correlates at approximately 0.97 with the simulated Southern-Central England data. Absolute interannual precipitation differences between the three data sets are at a maximum approximately 151 mm/season.

Page 14, Line 13: Suggest changing "relation" to "relationship".

Done

Page 14, Line 16: Suggest removing "though possibly not surprisingly".

Done

Page 14, Line 31: Suggest changing the part of the sentence after "period of" to "overlap".

Changed "the observational England-Wales time series" to "their overlap"

Page 14, Line 32: I think you mean "shorter timescale" here, "smaller scale" might be confusing since it could also refer to spatial scales.

We mean smaller amplitude and corrected this.

Page 15, Line 1: Suggest changing "lack the clear overall trend" to "do not have a long term trend".

Done

Page 15, Line 7: Suggest changing the part of the sentence after "indicate" to "the opposite".

Done

Page 15, Line 9: I would reference the most relevant figure for the "smoothed evolution" at the end of this sentence.

Done

Page 15, Line 13: Add "us" before "to".

Done

Page 15 Line 14: Why define an acronym for standard deviation here? If you wish to do so, it makes sense to do it at first mention in the methods section. You also do not use "SD" in the manuscript text, only in the figure labels—perhaps put the acronym definition in the figure caption.

We add it to the figure caption but also keep the note that SD in the figure refers to the Weibull Standard Deviation.

Page 15, Line 15: The Southern-Central reconstruction appears to have an increasing trend.

The reviewer is right. We amend this sentence: The reconstruction for East Anglia does not show a clear evolution in the Weibull standard deviations, whereas there is an increasing trend in the Weibull standard deviations for the Southern-Central England data.

Page 16, Line 12: It is worth being more specific here, something like "…do not agree on the time evolution of precipitation percentiles".

Done

Page 16, Line 12: There really does not seem to be much agreement. I think that this sentence confuses that point. Suggest reformatting to "Any agreement between the simulation and reconstructions appears random, with a tendency instead towards an opposite evolution—particularly for the dry percentile". You can then go into a bit more detail and the explanation in the following paragraph.

We change this to: Simulations and reconstructions do not agree on the time evolution of precipitation percentiles (Figures 4 to 6). Any hint of an agreement between reconstructed and simulated data is likely due to randomness (compare Figure 4). There is instead a tendency towards opposite time evolutions between the data sources. This is best seen in the dry percentiles from the mid-18th to mid-20th century (Figure 5).

Page 17, Line 3: Suggest changing to "…would show a range of trajectories and thus these results do not preclude that the model is capturing…"

We change this to: Obviously, using an ensemble of regional simulations would show a range of trajectories. Therefore, these results do not preclude per-se that the model is capturing basic physical characteristics of precipitation variability in northwestern Europe.

Page 17, Line 7: Suggest changing to "…more or less in opposition (Figure 3)."

Done

Figure 7 caption: I was not able to understand what this caption means.

We change this (and the following) to: Visualisation of how percentile-values change for over windows. We show, which percentile the 93.3th percentile MAMJJ precipitation amount for a reference window represents over time for England-Wales (green solid lines), Southern-Central England (blue dashed lines), and East Anglia (black dash-dotted lines) in a) reconstructions and observations, and b) simulations. The reference window is centred in 1815 CE.

Page 18, Line 7: Why 1815?

As stated on page 9, line 17 and now repeated in section 4.3: The year 1815 CE is included in all data sets and it allows equivalent analyses of the PMIP3 past1000 simulations (e.g., Schmidt et al., 2011).

Page 19, Line 14: Suggest removing "prior to approximately the year 1850 CE".

Changed as suggested.

Page 20, Line 11: I am not sure that I understand why this is specifically relevant to a comparison of precipitation and temperature. It seems more relevant to the comparisons of, for instance, model simulations to reconstructions.

Actually, our aim was to use this link to compare the different data sets. The statistical link is most likely a reflection of a physical mechanism, which ideally should be represented in all data sets.

We modify the subsection title to: Relation between Temperature and Precipitation in different data sources.

We modify the whole paragraph to read: We only shortly explore the interrelation between the regional temperature and precipitation variability. We show how interannual correlations between the precipitation records and temperature series evolve over time. Figure 10 plots sliding interannual correlations for 51-year windows between the observed and reconstructed precipitation data and the Central England temperature as well as the correlation between simulated England-Wales precipitation and simulated Central England temperature. We plot correlations for the untransformed precipitation records. All records are for the MAMJJ-season. Obviously, the high amount of internal variability on local and regional scales complicates the comparison among different data sources when studying small regions.

Page 20, Line 12: Suggest adding "thus" after "We" and removing "the" before "precipitation".

See reply to previous comment.

Page 22, Line 12: Add "the" before "environment".

Done

Page 22, Line 17: Suggest adding "inherent to SPI" after "standardization".

Added "inherent to the SPI"

Page 22, Line 21: Suggest removing the part of the sentence after "as well".

Done

Page 23, Line 16: Add "the" between "in" and "form".

Done. Also changed at two other instances.

Page 23, Line 17: Suggest putting a period after the Parker reference then starting a new sentence: "In addition, there are long instrumental station observations of precipitation…".

Done

Page 23, Line 18: I would remove these two sentences as there is not much value in discussing what you do not analyze.

OK. These were in response to earlier referee or editor comments.

Page 24, Line 5: Suggest moving "ACRE" into the parentheses and the full written out version into the main part of the sentence.

Done

Page 24, Line 10: You have already used PDSI at this point.

Thanks.

Sections 5.1 and 5.2: Much of these sections is redundant. I think the manuscript would be fine with all of this removed.

The structure of the discussion has been changed (see new section 5). These subsections have been removed here but a large part of the information was added to sections 2 and 3.

We do not completely remove these parts as they were included as response to concerns from the previous review-iteration.

Please see tracked changes in the discussions section, and earlier sections.

Page 27, Line 1: Suggest removing the period and reformatting as ", however, there is occasional temporal consistency between the simulations and observation based data."

This sentence and a good part of this paragraph was removed in response to general requests for shortening of the manuscript.

Page 27, Line 6 to 5.3.3: This is just more results and should either be moved to the results section or removed entirely.

We moved it to the results section.

Page 28, Line 3: Again this paragraph is largely results.

See previous response.

Page 28, Line 16: But it is really the reconstructions that look inconsistent with both the observations and simulations and thus I would focus predominantly on that in this paragraph (at the moment only the final sentence discusses this disagreement).

The paragraph changed to: The differences between simulation and observations may imply either shortcomings of any of the observational data sets in the early period or that the simulation presents a too stable relationship. Explanations might be physical inconsistencies within the simulations. More generally, any of the data sources may lack the physical relation between the temperature and precipitation records in the chosen season. Another possibility is that internal large-scale climate factors influencing the relation between both parameters evolve differently in simulation and reality. Assuming that the observations are the more reliable data set, we tend to the inference that the disagreement between observations and reconstructions suggest major shortcomings in the reconstructions set.

Page 28, Line 14: I would reformat this as "If we expect temporal consistency among the different sources of information, then we are assuming that all the sources of information are responding to the impact of external climate forcing".

Done

Page 28, Line 27: I would revisit this sentence with any eye towards clarity.

We rephrase: We have to ask, what is our expectation of consistency between simulated and observed responses to exogenous influences?

Page 28, Line 31: Change "frequent" to "frequency".

Done

Page 28, Line 34: Suggest changing to "However, the analysis period includes relatively large…"

Done

Page 29, Line 12: Does this not require that the opposite behavior is "significant" relative to randomness? This statement seems too speculative. Likewise, the observations and reconstructions often do not agree…

We remove the two sentences.

Page 30, Line 1: Change "frequent" to "frequency".

Done

Section 5.3.5: This discussion feels a bit too general and should be refocused towards discussing the results and implications of this study

We like to keep this but shorten it. Please refer to the new section 5.3 for details, e.g., in the tracked changes below.

**List of relevant changes:**

Abstract:

- Minor clarifications

Introduction:

- Minor clarifications
- Minor restructuring
- Overall possibly slightly shortened

Data:

- Added more information to Table 1
- Extended discussions on the individual sources of information
- Major clarifications and restructuring

Methods:

- Added more information
- Extended discussions of the method
- Major restructuring and clarifications

Results:

- Major clarifications
- Added additional analyses that were included in the Discussion section in the previous version

Discussions:

- Removed multiple subsections (content partly moved to other sections)
- Shortened remaining subsections
- Clarifications of the discussions

Conclusions:

- Clarifications and restructuring
- Highlighted main findings

Some references became obsolete.

[revised manuscript text omitted]

---

## Author Response (AR3)

Dear Jürg Luterbacher, dear reviewers,

please find attached the revised version of our manuscript considering your corrections.

Below we reply to your comments and summarise the made changes.

We would like to thank the anonymous reviewers and the editor Jürg Luterbacher for their help in improving the manuscript with their comments and suggestions as well as the editorial assistance.

On behalf of the authors,

Sincerely yours,

Oliver Bothe

Editor

dear authors, the comments of the Reviewers have arrived. As you can see, there are ony a few technical issues to be addressed.

best wishes and thanks

Jürg

Dear editor,

please find below our responses to the referees.

Thank you for your support in leading this manuscript to publication.

Best regards

Referee 2

The authors have addressed my comments to my satisfaction. I disagree that my remarks were radical, they reflected a question i was left with having read their paper. Addressing this point in in the discussion is useful.

Dear referee,

many thanks for your help.

Best regards

Referee 3

This is an update to a previously reviewed manuscript. The following comments are minor can be addressed without the need for another round of reviews.

Dear referee,

below we address your suggestions.

Thank you for your assistance in leading this manuscript to publication.

Best regards

Page 1, Line 10: Suggest changing "observational indices" to "observations".

We change this accordingly.

Page 1, Line 22: Suggest changing to "It strengthens our confidence if they prove to be consistent sources of information".

We change this to "It strengthens our confidence in inferrences if the sources of information prove to be consistent."

Page 2, Line 6: Suggest changing "data fields, the grid resolution" to "observations, the spatial resolution".

We change this to "fields of data, the spatial resolution".

Page 3, Line 3: Suggest changing "the South of Great Britain" to "southern Great Britain" to be consistent with the more common usage in the manuscript.

We change this accordingly.

Page 3, Line 9: Suggest removing "a simulation" after the comma.

We do so.

Page 3, Line 23: Suggest changing "not least" to "also".

We change this.

Page 3, Line 26: Suggest moving the comma to before "especially".

We do so.

Page 3, Line 29: Suggest removing "compare".

We remove this.

Page 4, Line 1: Suggest changing "on" to "of".

We change this.

Page 4, Line 10: Suggest changing "by" to "from".

We change this.

Page 5, Line 20: Suggest changing "the South of Great Britain" to "southern Great Britain" to be consistent with the more common usage in the manuscript.

We do so.

Page 5, Line 33: Suggest changing "are, e.g.," to "include".

We change this.

Page 6, Line 2: Suggest changing "allow assessing" to "allow for assessment of".

We do so.

Page 6, Line 7: Suggest changing "the South of Great Britain" to "southern Great Britain" to be consistent with the more common usage in the manuscript.

We do so.

Page 7, Line 7: Suggest putting June to August in parentheses and an "a" before "scaling" and "a" on the following line before "supplement".

We follow all three suggestions.

Page 7, Line 12: Add the year of the reference in parentheses.

We changed the reference style.

Page 7, Line 23: Suggest removing the sentence starting "However" as it does not appear to add useful information.

We remove the sentence.

Page 7, Line 26: Suggest changing "the South of Britain" to "southern Great Britain" to be consistent with the more common usage in the manuscript.

We do so.

Page 7, Line 30: Suggest starting the sentence "To our knowledge, this simulation…" and removing the part after the comma at the end of the sentence.

We change this accordingly.

Page 8, Line 13: Suggest changing "degree" to "degrees", also on Line 15.

We change both instances.

Page 9, Line 12: Suggest removing "at least".

We do so.

Page 9, Line 16: Suggest changing "on" to "when".

We do so.

Page 10, Line 11: Suggest combining these two sentences "data and there are…".

We do so and, then, also combine this paragraph with the following one.

Page 10, Line 25: Suggest removing "at least".

As the recommendation by McKee et al. is explicitly for "at least 30" data points, we would prefer to keep this here.

Page 10, Line 29: Suggest changing "this" to "the" and removing "interannually" on the following line.

We apply both suggestions.

Page 11, Line 5: Suggest adding "can" before "point".

We do so.

Page 11, Line 5: Suggest changing "calculating" to "for the calculation of".

We change this.

Page 11, Line 8: Suggest flipping "represent traditionally".

We do so.

Page 11, Line 15: Suggest adding "the" before "reference".

We do so.

Page 11, Line 30: I might add at the end of the sentence: "which have too much high-frequency variability and are too-smoothed, respectively".

We add: " which have too much high-frequency variability or are too-smoothed, respectively"

Figure 2 caption: Suggest removing the comma in the last sentence and replace it with "that".

We do so.

Page 13, Line 7: Suggest changing the end of the sentence to: "than the agreement amongst the observational data".

We do so, but are unsure, which phrasing is better.

Page 13, Line 10: Suggest removing "on interannual time-scales".

We do so.

Page 13, Line 16: Suggest changing the beginning of the sentence to "There is a strong relationship...".

Since page 13 has not a Line 16, we are unclear, whether the referee refers to page 13 Line 13 or page 14 Line 1. Therefore, we do not follow this suggestion.

Page 14, Line 1: Suggest changing "relation" to "relationship" here and on Lines 3 and 7.

We do so for the first two instances but prefer "Temperature-relations" on Line 7.

Page 14, Line 12: Suggest removing "for this resolution".

We do so.

Page 17, Line 4: Suggest changing "93.3th, i.e. wet, percentile" to "wet percentile" and a similar change with the dry percentile statement on Line 8.

We follow both suggestions.

Page 17, Line 10: Suggest adding "before transformation into a distributional form" at the end of the sentence.

We do so.

Page 19, Line 9: Suggest removing "per-se".

We do so.

Page 20, Line 8: Remove "each window".

We thank the referee for spotting this, and we remove it.

Page 21, Line 2: Suggest changing "of" to "for".

We do so.

Page 21, Line 6: Suggest adding "the" before the list of percentiles.

We do so.

Page 22, Line 4: Suggest removing "shortly".

We remove shortly in Line 1 of page 22.

Page 22, Line 8: I understand what you are saying but the sentence is a bit confusing, I suggest you revisit this with an eye towards clarity.

We change this to: The amount of precipitation, which represents median values for the reference year 1815 CE, is representative of larger percentiles in later years (Figure \ref{fig:fig9}b). However, there is a slight decreasing trend from approximately the mid-19th century to the end of the simulation (Figure \ref{fig:fig9}b).

Page 23, Line 11: Suggest changing "show again" to "again show".

We change this.

Page 23, Line 15: Suggest changing "only shortly" to "briefly". I would also move up the first sentence of the next paragraph into this first paragraph, changing the beginning of that sentence to read "In particular, we show...for this season.".

We follow the suggestions.

Page 24, Line 2: Suggest changing "high" to "large" and adding "such" before "small".

We follow these suggestions.

Page 24, Line 9: Suggest combining these two sentences.

It is unclear which sentences are meant. We combine the sentences on Line 10: We do not show significance levels in Figure \ref{fig:fig11} but we note that for 51-year windows and the time series characteristics of the data (e.g., approximately uncorrelated noise for seasonal precipitation), one may regard absolute values of correlation coefficients larger than \(0.23\) as statistically significant at the 5% level.

Page 24, Line 17: Suggest changing "relation" to "relationship".

We do so.

Page 24, Line 24: Suggest changing "shortly" to "briefly".

We do so.

Page 24, Line 27: Suggest changing "simulations" to "simulated".

We change this.

Page 24, Line 4: Suggest changing "relation" to "relationship".

We change this.

Page 25, Line 17: Suggest changing "shows again" to "again shows".

We change this.

Page 25, Line 21: Suggest changing "in" to "with".

We change "in" to "by showing"

Page 25, Line 22: Suggest changing "relation" to "relationship".

We do so.

Page 25, Line 26: Suggest removing "of climate parameters".

We remove this.

Page 25, Line 32: Suggest changing "which again" to "that" and removing "the input changed over time". It is unclear what that sentence means—if you choose not to remove it you should revisit the sentence with an eye towards clarity.

We modify the two sentences to: The observational England-Wales precipitation data is a weighted composite of regional series based on instrumental information. The information entering the composites and the regional index changed over time.

Page 25, Line. 33: Suggest changing "proxy" to "proxies".

We do so.

Page 26, Line 8: Suggest changing "relation" to "relationship".

We do so.

Page 26, Line 11: Suggest removing "a priori" and "on" to "as to" on the following line.

We do so.

Page 26, Line 19: Suggest changing "to identify" to "the identification of".

We do so.

Page 26, Line 29: Suggest changing "differences to" to "differences as compared to".

We change this accordingly.

Page 26, Line 32: Suggest changing "the reconstructions relation to temperature" to "the relationship between the reconstructions and temperature" and "relate" to "reflect" on the following line.

We change this.

Page 27, Line 6: Suggest changing "relations" to "relationships" and removing "from March to July". I would also change "relation" to "relationship" throughout this paragraph and the next.

We make all these changes.

Page 27, Line 14: Suggest adding "the" before "simulation". I would also clarify what relationship you are talking about on the following line.

We add "the" and add at the end of the sentence " between temperature and precipitation in southern Great Britain".

Page 27, Line 18: Suggest adding "the" before "simulation".

We do so.

Page 27, Line 24: Suggest adding "magnitude" after "large".

We add "amplitude" after large.

Page 27, Line 27: Suggest removing "large".

We do so.

Page 28, Line 9: Suggest adding "variability in" before "the North…"

We do so.

Page 28, Line 19: What is meant by "increase our confidence in forced changes".

We add "simulated" before "forced changes".

Page 28, Line 25: Suggest changing "comparing" to "comparison of".

We change this accordingly.

Page 28, Line 30: Suggest removing "indeed".

We do so.

Page 28, Line 32: Suggest changing "Particularly" to "In particular,". This sentence does not appear to be consistent with the previous sentence where you note that they have limited agreement. "Comparable changes" implies consistency while "limited agreement" implies inconsistency.

We change both sentences to: Second, the regional simulation shows occasional agreement with its observational target, the observational England-Wales precipitation data. In particular the variability in both data sources shows comparable changes for the full period of the observations.

Page 28, Line 33: Suggest removing "mainly".

We do so.

Page 29, Line 2: This sentence is vague, I suggest that you expand on the "different processes".

We change the sentence to: However, considering all associated uncertainties, we can not conclude that the agreement in properties does reflect agreement in the underlying processes in the respective data sources.

Page 29, Line 8: Suggest changing "relation" to "relationship" here and on the next line.

We change these two occurrences.

List of changes

Clarifications and corrections according to referee 3's suggestions.

Extended acknowledgements.

Grammar and spelling corrections in the supplement.

[revised manuscript text omitted]